# Contextual Dropout: An Efficient Sample-Dependent Dropout Module

**Xinjie Fan**[*,1]**, Shujian Zhang**[*,1]**, Korawat Tanwisuth**[1]**, Xiaoning Qian**[2]**, Mingyuan Zhou**[1]
[1]The University of Texas at Austin, [2]Texas A&M University,
`xfan@utexas.edu, szhang19@utexas.edu,`
`korawat.tanwisuth@utexas.edu,xqian@ece.tamu.edu,`
`mingyuan.zhou@mccombs.utexas.edu`

## Abstract

Dropout has been demonstrated as a simple and effective module to not only regularize the training process of deep neural networks, but also provide the uncertainty estimation for prediction. However, the quality of uncertainty estimation is highly dependent on the dropout probabilities. Most current models use the same dropout distributions across all data samples due to its simplicity. Despite the potential gains in the flexibility of modeling uncertainty, sample-dependent dropout, on the other hand, is less explored as it often encounters scalability issues or involves non-trivial model changes. In this paper, we propose contextual dropout with an efficient structural design as a simple and scalable sample-dependent dropout module, which can be applied to a wide range of models at the expense of only slightly increased memory and computational cost. We learn the dropout probabilities with a variational objective, compatible with both Bernoulli dropout and Gaussian dropout. We apply the contextual dropout module to various models with applications to image classification and visual question answering and demonstrate the scalability of the method with large-scale datasets, such as ImageNet and VQA 2.0. Our experimental results show that the proposed method outperforms baseline methods in terms of both accuracy and quality of uncertainty estimation.

## 1 Introduction

Deep neural networks (NNs) have become ubiquitous and achieved state-of-the-art results in a wide variety of research problems (LeCun et al., 2015). To prevent over-parameterized NNs from overfitting, we often need to appropriately regularize their training. One way to do so is to use Bayesian NNs that treat the NN weights as random variables and regularize them with appropriate prior distributions (MacKay, 1992; Neal, 2012). More importantly, we can obtain the model's confidence on its predictions by evaluating the consistency between the predictions that are conditioned on different posterior samples of the NN weights. However, despite significant recent efforts in developing various types of approximate inference for Bayesian NNs (Graves, 2011; Welling & Teh, 2011; Li et al., 2016; Blundell et al., 2015; Louizos & Welling, 2017; Shi et al., 2018), the large number of NN weights makes it difficult to scale to real-world applications.

Dropout has been demonstrated as another effective regularization strategy, which can be viewed as imposing a distribution over the NN weights (Gal & Ghahramani, 2016). Relating dropout to Bayesian inference provides a much simpler and more efficient way than using vanilla Bayesian NNs to provide uncertainty estimation (Gal & Ghahramani, 2016), as there is no more need to explicitly instantiate multiple sets of NN weights. For example, Bernoulli dropout randomly shuts down neurons during training (Hinton et al., 2012; Srivastava et al., 2014). Gaussian dropout multiplies the neurons with independent, and identically distributed ($iid$) Gaussian random variables drawn from $\mathcal{N}(1, \alpha)$, where the variance $\alpha$ is a tuning parameter (Srivastava et al., 2014). Variational dropout generalizes Gaussian dropout by reformulating it under a Bayesian setting and allowing $\alpha$ to be learned under a variational objective (Kingma et al., 2015; Molchanov et al., 2017).

---

* Equal contribution. Corresponding to: `mingyuan.zhou@mccombs.utexas.edu`

However, the quality of uncertainty estimation depends heavily on the dropout probabilities (Gal et al., 2017). To avoid grid-search over the dropout probabilities, Gal et al. (2017) and Boluki et al. (2020) propose to automatically learn the dropout probabilities, which not only leads to a faster experiment cycle but also enables the model to have different dropout probabilities for each layer, bringing greater flexibility into uncertainty modeling. But, these methods still impose the restrictive assumption that dropout probabilities are global parameters shared across all data samples. By contrast, we consider parameterizing dropout probabilities as a function of input covariates, treating them as data-dependent local variables. Applying covariate-dependent dropouts allows different data to have different distributions over the NN weights. This generalization has the potential to greatly enhance the expressiveness of a Bayesian NN. However, learning covariate-dependent dropout rates is challenging. Ba & Frey (2013) propose *standout*, where a binary belief network is laid over the original network, and develop a heuristic approximation to optimize free energy. But, as pointed out by Gal et al. (2017), it is not scalable due to its need to significantly increase the model size.

In this paper, we propose a simple and scalable contextual dropout module, whose dropout rates depend on the covariates $x$, as a new approximate Bayesian inference method for NNs. With a novel design that reuses the main network to define how the covariate-dependent dropout rates are produced, it boosts the performance while only slightly increases the memory and computational cost. Our method greatly enhances the flexibility of modeling, maintains the inherent advantages of dropout over conventional Bayesian NNs, and is generally simple to implement and scalable to the large-scale applications. We plug the contextual dropout module into various types of NN layers, including fully connected, convolutional, and attention layers. On a variety of supervised learning tasks, contextual dropout achieves good performance in terms of accuracy and quality of uncertainty estimation.

## 2 CONTEXTUAL DROPOUT

We introduce an efficient solution for data-dependent dropout: (1) treat the dropout probabilities as sample-dependent local random variables, (2) propose an efficient parameterization of dropout probabilities by sharing parameters between the encoder and decoder, and (3) learn the dropout distribution with a variational objective.

### 2.1 BACKGROUND ON DROPOUT MODULES

Consider a supervised learning problem with training data $\mathcal{D} := \{x_i, y_i\}_{i=1}^N$, where we model the conditional probability $p_\theta(y_i \,|\, x_i)$ using a NN parameterized by $\theta$. Applying dropout to a NN often means element-wisely reweighing each layer with a data-specific Bernoulli/Gaussian distributed random mask $z_i$, which are $iid$ drawn from a prior $p_\eta(z)$ parameterized by $\eta$ (Hinton et al., 2012; Srivastava et al., 2014). This implies dropout training can be viewed as approximate Bayesian inference (Gal & Ghahramani, 2016). More specifically, one may view the learning objective of a supervised learning model with dropout as a log-marginal-likelihood: $\log \int \prod_{i=1}^N p(y_i \,|\, x_i, z)p(z)dz$. To maximize this often intractable log-marginal, it is common to resort to variational inference (Hoffman et al., 2013; Blei et al., 2017) that introduces a variational distribution $q(z)$ on the random mask $z$ and optimizes an evidence lower bound (ELBO):

$$\mathcal{L}(\mathcal{D}) = \mathbb{E}_{q(z)} \left[ \log \frac{\prod_{i=1}^N p_\theta(y_i \,|\, x_i, z)p_\eta(z)}{q(z)} \right] = \left( \textstyle\sum_{i=1}^N \mathbb{E}_{z_i \sim q(z)} \left[ \log p_\theta(y_i \,|\, x_i, z_i) \right] \right) - \mathrm{KL}(q(z)||p_\eta(z)), \quad (1)$$

where $\mathrm{KL}(q(z)||p_\eta(z)) = \mathbb{E}_{q(z)}[\log q(z) - \log p(z)]$ is a Kullback–Leibler (KL) divergence based regularization term. Whether the KL term is explicitly imposed is a key distinction between regular dropout (Hinton et al., 2012; Srivastava et al., 2014) and their Bayesian generalizations (Gal & Ghahramani, 2016; Gal et al., 2017; Kingma et al., 2015; Molchanov et al., 2017; Boluki et al., 2020).

### 2.2 COVARIATE-DEPENDENT WEIGHT UNCERTAINTY

In regular dropout, as shown in (1), while we make the dropout masks data specific during optimization, we keep their distributions the same. This implies that while the NN weights can vary from data to data, their distribution is kept data invariant. In this paper, we propose *contextual dropout*, in which the distributions of dropout masks $z_i$ depend on covariates $x_i$ for each sample $(x_i, y_i)$. Specifically, we define the variational distribution as $q_\phi(z_i \,|\, x_i)$, where $\phi$ denotes its NN parameters. In the framework of amortized variational Bayes (Kingma & Welling, 2013; Rezende

et al., 2014), we can view $q_\phi$ as an inference network (encoder) trying to approximate the posterior $p(\boldsymbol{z}_i \,|\, y_i, \boldsymbol{x}_i) \propto p(y_i \,|\, \boldsymbol{x}_i, \boldsymbol{z}_i)p(\boldsymbol{z}_i)$. Note as we have no access to $y_i$ during testing, we parameterize our encoder in a way that it depends on $\boldsymbol{x}_i$ but not $y_i$. From the optimization point of view, what we propose corresponds to the ELBO of $\log \prod_{i=1}^{N} \int p(y_i \,|\, \boldsymbol{x}_i, \boldsymbol{z}_i)p(\boldsymbol{z}_i)d\boldsymbol{z}_i$ given $q_\phi(\boldsymbol{z}_i \,|\, \boldsymbol{x}_i)$ as the encoder, which can be expressed as

$$\mathcal{L}(\mathcal{D}) = \sum_{i=1}^{N} \mathcal{L}(\boldsymbol{x}_i, y_i), \ \mathcal{L}(\boldsymbol{x}_i, y_i) = \mathbb{E}_{\boldsymbol{z}_i \sim q_\phi(\cdot \,|\, \boldsymbol{x}_i)}[\log p_\theta(y_i \,|\, \boldsymbol{x}_i, \boldsymbol{z}_i)] - \mathrm{KL}(q_\phi(\boldsymbol{z}_i \,|\, \boldsymbol{x}_i) || p_\eta(\boldsymbol{z}_i)). \quad (2)$$

This ELBO differs from that of regular dropout in (1) in that the dropout distributions for $\boldsymbol{z}_i$ are now parameterized by $\boldsymbol{x}_i$ and a single KL regularization term is replaced with the aggregation of $N$ data-dependent KL terms. Unlike conventional Bayesian NNs, as $\boldsymbol{z}_i$ is now a local random variable, the impact of the KL terms will not diminish as $N$ increases, and from the viewpoint of uncertainty quantification, contextual dropout relies only on aleatoric uncertainty to model its uncertainty on $y_i$ given $\boldsymbol{x}_i$. Like conventional BNNs, we may add epistemic uncertainty by imposing a prior distribution on $\theta$ and/or $\phi$, and infer their posterior given $\mathcal{D}$. As contextual dropout with a point estimate on both $\theta$ and $\phi$ is already achieving state-of-the-art performance, we leave that extension for future research. In what follows, we omit the data index $i$ for simplification and formally define its model structure.

**Cross-layer dependence:** For a NN with $L$ layers, we denote $\boldsymbol{z} = \{\boldsymbol{z}^1, \ldots, \boldsymbol{z}^L\}$, with $\boldsymbol{z}^l$ representing the dropout masks at layer $l$. As we expect $\boldsymbol{z}^l$ to be dependent on the dropout masks in previous layers $\{\boldsymbol{z}^j\}_{j<l}$, we introduce an autoregressive distribution as $q_\phi(\boldsymbol{z} \,|\, \boldsymbol{x}) = \prod_{l=1}^{L} q_\phi(\boldsymbol{z}^l \,|\, \boldsymbol{x}^{l-1})$, where $\boldsymbol{x}^{l-1}$, the output of layer $l-1$, is a function of $\{\boldsymbol{z}^1, \ldots, \boldsymbol{z}^{l-1}, \boldsymbol{x}\}$.

**Parameter sharing between encoder and decoder:** We aim to build an encoder by modeling $q_\phi(\boldsymbol{z}^l \,|\, \boldsymbol{x}^{l-1})$, where $\boldsymbol{x}$ may come from complex and highly structured data such as images and natural languages. Thus, extracting useful features from $\boldsymbol{x}$ to learn the encoder distribution $q_\phi$ itself becomes a problem as challenging as the original one, *i.e.*, extracting discriminative features from $\boldsymbol{x}$ to predict $y$. As intermediate layers in the decoder network $p_\theta$ are already learning useful features from the input, we choose to reuse them in the encoder, instead of extracting the features from scratch. If we denote layer $l$ of the decoder network by $g_\theta^l$, then the output of layer $l$, given its input $\boldsymbol{x}^{l-1}$, would be $\mathbf{U}^l = g_\theta^l(\boldsymbol{x}^{l-1})$. Considering this as a learned feature for $\boldsymbol{x}$, as illustrated in Figure 1, we build the encoder on this output as

$\boldsymbol{\alpha}^l = h_\varphi^l(\mathbf{U}^l)$, draw $\boldsymbol{z}^l$ conditioning on $\boldsymbol{\alpha}^l$, and element-wisely multiply $\boldsymbol{z}^l$ with $\mathbf{U}^l$ (with broadcast if needed) to produce the output of layer $l$ as $\boldsymbol{x}^l$. In this way, we use $\{\theta, \varphi\}$ to parameterize the encoder, which reuses parameters $\theta$ of the decoder. To produce the dropout rates of the encoder, we only need extra parameters $\varphi$, the added memory and computational cost of which are often insignificant in comparison to these of the decoder.

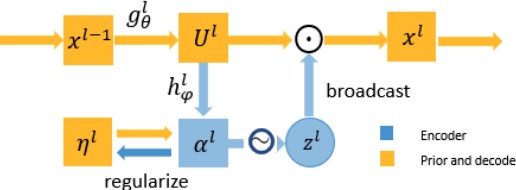

Figure 1: A contextual dropout module.

## 2.3 Efficient parameterization of contextual dropout module

Denote the output of layer $l$ by a multidimensional array (tensor) $\mathbf{U}^l = g_\theta^l(\boldsymbol{x}^{l-1}) \in \mathbb{R}^{C_1^l \times \ldots \times C_{D^l}^l}$, where $D^l$ denotes the number of the dimensions of $\mathbf{U}^l$ and $C_d^l$ denotes the number of elements along dimension $d \in \{1, \ldots, D^l\}$. For efficiency, the output shape of $h_\varphi^l$ is not matched to the shape of $\mathbf{U}^l$. Instead, we make it smaller and broadcast the contextual dropout masks $\boldsymbol{z}^l$ across the dimensions of $\mathbf{U}^l$ (Tompson et al., 2015). Specifically, we parameterize dropout logits $\boldsymbol{\alpha}^l$ of the variational distribution to have $C_d^l$ elements, where $d \in \{1, \ldots, D^l\}$ is a specified dimension of $\mathbf{U}^l$. We sample $\boldsymbol{z}^l$ from the encoder and broadcast them across all but dimension $d$ of $\mathbf{U}^l$. We sample $\boldsymbol{z}^l \sim \mathrm{Ber}(\sigma(\boldsymbol{\alpha}^l))$ under contextual Bernoulli dropout, and follow Srivastava et al. (2014) to use $\boldsymbol{z}^l \sim N(1, \sigma(\boldsymbol{\alpha}^l)/(1 - \sigma(\boldsymbol{\alpha}^l)))$ for contextual Gaussian dropout. To obtain $\boldsymbol{\alpha}^l \in \mathbb{R}^{C_d^l}$, we first take the average pooling of $\mathbf{U}^l$ across all but dimension $d$, with the output denoted as $F_{\text{avepool},d}(\mathbf{U}^l)$, and then apply two fully-connected layers $\Phi_1^l$ and $\Phi_2^l$ connected by $F_{\text{NL}}$, a (Leaky) ReLU based nonlinear activation function, as

$$\boldsymbol{\alpha}^l = h_\varphi^l(\mathbf{U}^l) = \Phi_2^l(F_{\text{NL}}(\Phi_1^l(F_{\text{avepool},d}(\mathbf{U}^l)))), \quad (3)$$

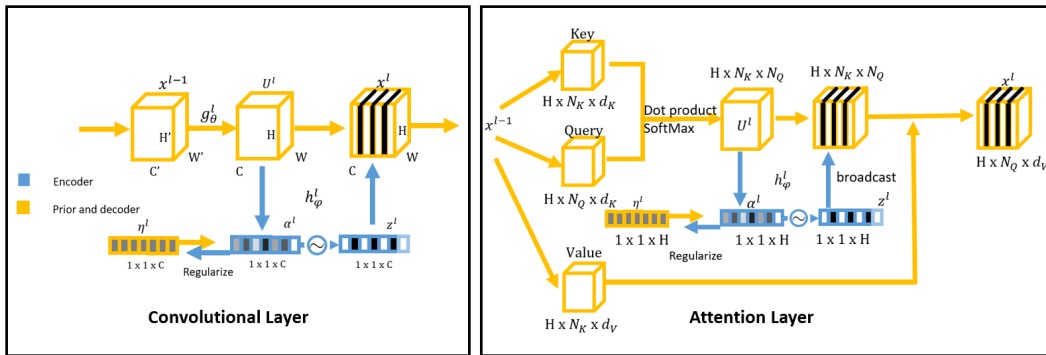

Figure 2: Left: Contextual dropout in convolution layers. Right: Contextual dropout in attention layers.

where $\Phi_1^l$ is a linear transformation mapping from $\mathbb{R}^{C_d^l}$ to $\mathbb{R}^{C_d^l/\gamma}$, while $\Phi_2^l$ is from $\mathbb{R}^{C_d^l/\gamma}$ back to $\mathbb{R}^{C_d^l}$, with $\gamma$ being a reduction ratio controlling the complexity of $h_{\boldsymbol{\varphi}}^l$. Below we describe how to apply contextual dropout to three representative types of NN layers.

*Contextual dropout module for fully-connected layers[2]:* If layer $l$ is a fully-connected layer and $\mathbf{U}^l \in \mathbb{R}^{C_1^l \times \cdots \times C_{D^l}^l}$, we set $\boldsymbol{\alpha}^l \in \mathbb{R}^{C_{D^l}^l}$, where $D^l$ is the dimension that the linear transformation is applied to. Note, if $\mathbf{U}^l \in \mathbb{R}^{C_1^l}$, then $\boldsymbol{\alpha}^l \in \mathbb{R}^{C_1^l}$, and $F_{\text{avepool},1}$ is an identity map, so $\boldsymbol{\alpha}^l = \Phi_2^l(F_{\text{NL}}(\Phi_1^l(\mathbf{U}^l)))$.

*Contextual dropout module for convolutional layers:* Assume layer $l$ is a convolutional layer with $C_3^l$ as convolutional channels and $\mathbf{U}^l \in \mathbb{R}^{C_1^l \times C_2^l \times C_3^l}$. Similar to Spatial Dropout (Tompson et al., 2015), we set $\boldsymbol{\alpha}^l \in \mathbb{R}^{C_3^l}$ and broadcast its corresponding $\boldsymbol{z}^l$ spatially as illustrated in Figure 2. Such parameterization is similar to the squeeze-and-excitation unit for convolutional layers, which has been shown to be effective in image classification tasks (Hu et al., 2018). However, in squeeze-and-excitation, $\sigma(\boldsymbol{\alpha}^l)$ is used as channel-wise soft attention weights instead of dropout probabilities, therefore it serves as a deterministic mapping in the model instead of a stochastic unit used in the inference network.

*Contextual dropout module for attention layers:* Dropout has been widely used in attention layers (Xu et al., 2015b; Vaswani et al., 2017; Yu et al., 2019). For example, it can be applied to multi-head attention weights after the softmax operation (see illustrations in Figure 2). The weights are of dimension $[H, N_K, N_Q]$, where $H$ is the number of heads, $N_K$ the number of keys, and $N_Q$ the number of queries. In this case, we find that setting $\boldsymbol{\alpha}^l \in \mathbb{R}^H$ gives good performance. Intuitively, this coincides with the choice of channel dimension for convolutional layers, as heads in attention could be analogized as channels in convolution.

## 2.4 VARIATIONAL INFERENCE FOR CONTEXTUAL DROPOUT

In contextual dropout, we choose $\mathcal{L}(\mathcal{D}) = \sum_{(\boldsymbol{x},y)\in\mathcal{D}} \mathcal{L}(\boldsymbol{x}, y)$ shown in (2) as the optimization objective. Note in our design, the encoder $q_{\boldsymbol{\phi}}$ reuses the decoder parameters $\boldsymbol{\theta}$ to define its own parameters. Therefore, we copy the values of $\boldsymbol{\theta}$ into $\boldsymbol{\phi}$ and stop the gradient of $\boldsymbol{\theta}$ when optimizing $q_{\boldsymbol{\phi}}$. This is theoretically sound (Ba & Frey, 2013). Intuitively, the gradients to $\boldsymbol{\theta}$ from $p_{\boldsymbol{\theta}}$ are less noisy than that from $q_{\boldsymbol{\phi}}$ as the training of $p_{\boldsymbol{\theta}}(y \,|\, \boldsymbol{x}, \boldsymbol{z})$ is supervised while that of $q_{\boldsymbol{\phi}}(\boldsymbol{z})$ is unsupervised. As what we have expected, allowing gradients from $q_{\boldsymbol{\phi}}$ to backpropagate to $\boldsymbol{\theta}$ is found to adversely affect the training of $p_{\boldsymbol{\theta}}$ in our experiments. We use a simple prior $p_{\boldsymbol{\eta}}$, making the prior distributions for dropout masks the same within each layer. The gradients with respect to $\boldsymbol{\eta}$ and $\boldsymbol{\theta}$ can be expressed as

$$\nabla_{\boldsymbol{\eta}}\mathcal{L}(\boldsymbol{x},y) = \mathbb{E}_{\boldsymbol{z}\sim q_{\boldsymbol{\phi}}(\cdot\,|\,\boldsymbol{x})}[\nabla_{\boldsymbol{\eta}}\log p_{\boldsymbol{\eta}}(\boldsymbol{z})], \quad \nabla_{\boldsymbol{\theta}}\mathcal{L}(\boldsymbol{x},y) = \mathbb{E}_{\boldsymbol{z}\sim q_{\boldsymbol{\phi}}(\cdot\,|\,\boldsymbol{x})}[\nabla_{\boldsymbol{\theta}}\log p_{\boldsymbol{\theta}}(y\,|\,\boldsymbol{x},\boldsymbol{z})], \quad (4)$$

which are both estimated via Monte Carlo integration, using a single $\boldsymbol{z} \sim q_{\boldsymbol{\phi}}(\boldsymbol{z}\,|\,\boldsymbol{x})$ for each $\boldsymbol{x}$.

Now, we consider the gradient of $\mathcal{L}$ with respect to $\boldsymbol{\varphi}$, the components of $\boldsymbol{\phi} = \{\boldsymbol{\theta}, \boldsymbol{\varphi}\}$ not copied from the decoder. For Gaussian contextual dropout, we estimate the gradients via the reparameterization

---

[2]Note that full-connected layers can be applied to multi-dimensional tensor as long as we specify the dimension along which the summation operation is conducted (Abadi et al., 2015).

trick (Kingma & Welling, 2013). For $z^l \sim N(1, \sigma(\alpha^l)/(1 - \sigma(\alpha^l)))$, we rewrite it as $z^l = 1 + \sqrt{\sigma(\alpha^l)/(1 - \sigma(\alpha^l))}\epsilon^l$, where $\epsilon^l \sim \mathcal{N}(0, I)$. Similarly, sampling a sequence of $z = \{z^l\}_{l=1}^L$ from $q_\phi(z \mid x)$ can be rewritten as $f_\phi(\epsilon, x)$, where $f_\phi$ is a deterministic differentiable mapping and $\epsilon$ are $iid$ standard Gaussian. The gradient $\nabla_\varphi \mathcal{L}(x, y)$ can now be expressed as (see pseudo code of Algorithm 3 in Appendix)

$$\nabla_\varphi \mathcal{L}(x, y) = \mathbb{E}_{\epsilon \sim \mathcal{N}(0,1)}[\nabla_\varphi(\log p_\theta(y \mid x, f_\phi(\epsilon, x)) - \tfrac{\log q_\phi(f_\phi(\epsilon, x) \mid x)}{\log p_\eta(f_\phi(\epsilon, x))})]. \tag{5}$$

For Bernoulli contextual dropout, backpropagating the gradient efficiently is not straightforward, as the Bernoulli distribution is not reparameterizable, restricting the use of the reparameterization trick. In this case, a commonly used gradient estimator is the REINFORCE estimator (Williams, 1992) (see details in Appendix A). This estimator, however, is known to have high Monte Carlo estimation variance. To this end, we estimate $\nabla_\varphi \mathcal{L}$ with the augment-REINFORCE-merge (ARM) estimator (Yin & Zhou, 2018), which provides unbiased and low-variance gradients for the parameters of Bernoulli distributions. We defer the details of this estimator to Appendix A. We note there exists an improved ARM estimator (Yin et al., 2020; Dong et al., 2020), applying which could further improve the performance.

## 2.5 TESTING AND COMPLEXITY ANALYSIS

**Testing stage:** To obtain a point estimate, we follow the common practice in dropout (Srivastava et al., 2014) to multiply the neurons by the expected values of random dropout masks, which means that we predict $y$ with $p_\theta(y \mid x, \bar{z})$, where $\bar{z} = \mathbb{E}_{q_\phi(z \mid x)}[z]$ under the proposed contextual dropout. When uncertainty estimation is needed, we draw $K$ random dropout masks to approximate the posterior predictive distribution of $y$ given $x$ using $\hat{p}(y \mid x) = \frac{1}{K} \sum_{k=1}^K p_\theta(y \mid x, z^{(k)})$, where $z^{(1)}, \ldots, z^{(K)} \overset{iid}{\sim} q_\phi(z \mid x)$.

**Complexity analysis:** The added computation and memory of contextual dropout are insignificant due to the parameter sharing between the encoder and decoder. Extra memory and computational cost mainly comes from the part of $h_\varphi^l$, where both the parameter size and number of operations are of order $O((C_d^l)^2/\gamma)$, where $\gamma$ is from 8 to 16. This is insignificant, compared to the memory and computational cost of the main network, which are of order larger than $O((C_d^l)^2)$. We verify the point by providing memory and runtime comparisons between contextual dropout and other dropouts on ResNet in Table 3 (see more model size comparisons in Table 5 in Appendix).

## 2.6 RELATED WORK

*Data-dependent variational distribution:* Deng et al. (2018) model attentions as latent-alignment variables and optimize a tighter lower bound (compared to hard attention) using a learned inference network. To balance exploration and exploitation for contextual bandits problems, Wang & Zhou (2019) introduce local variable uncertainty under the Thompson sampling framework. However, their inference networks of are both independent of the decoder, which may considerably increase memory and computational cost for the considered applications. Fan et al. (2020) propose Bayesian attention modules with efficient parameter sharing between the encoder and decoder networks. Its scope is limited to attention units as Deng et al. (2018), while we demonstrate the general applicability of contextual dropout to fully connected, convolutional, and attention layers in supervised learning models. *Conditional computation* (Bengio et al., 2015; 2013; Shazeer et al., 2017; Teja Mullapudi et al., 2018) tries to increase model capacity without a proportional increase in computation, where an independent gating network decides turning which part of a network active and which inactive for each example. In contextual dropout, the encoder works much like a gating network choosing the distribution of sub-networks for each sample. But the potential gain in model capacity is even larger, *e.g.*, there are potentially $\sim O((2^d)^L)$ combinations of nodes for $L$ fully-connected layers, where $d$ is the order of the number of nodes for one layer. *Generalization of dropout*: DropConnect (Wan et al., 2013) randomly drops the weights rather than the activations so as to generalize dropout. The dropout distributions for the weights, however, are still the same across different samples. Contextual dropout utilizes sample-dependent dropout probabilities, allowing different samples to have different dropout probabilities.

## 3 EXPERIMENTS

Our method can be straightforwardly deployed wherever regular dropout can be utilized. To test its general applicability and scalability, we apply the proposed method to three representative types of NN layers: fully connected, convolutional, and attention layers with applications on MNIST (LeCun et al., 2010), CIFAR (Krizhevsky et al., 2009), ImageNet (Deng et al., 2009), and VQA-v2 (Goyal et al., 2017). To investigate the model's robustness to noise, we also construct noisy versions of datasets by adding Gaussian noises to image inputs (Larochelle et al., 2007).

For evaluation, we consider both the accuracy and uncertainty on predicting $y$ given $x$. Many metrics have been proposed to evaluate the quality of uncertainty estimation. On one hand, researchers are generating calibrated probability estimates to measure model confidence (Guo et al., 2017; Naeini et al., 2015; Kuleshov et al., 2018). While expected calibration error and maximum calibration error have been proposed to quantitatively measure calibration, such metrics do not reflect how robust the probabilities are with noise injected into the network input, and cannot capture epistemic or model uncertainty (Gal & Ghahramani, 2016). On the other hand, the entropy of the predictive distribution as well as the mutual information, between the predictive distribution and posterior over network weights, are used as metrics to capture both epistemic and aleatoric uncertainty (Mukhoti & Gal, 2018). However, it is often unclear how large the entropy or mutual information is large enough to be classified as uncertain, so such metric only provides a relative uncertainty measure.

**Hypothesis testing based uncertainty estimation**: Unlike previous information theoretic metrics, we use a statistical test based method to estimate uncertainty, which works for both single-label and multi-label classification models. One advantage of using hypothesis testing over information theoretic metrics is that the $p$-value of the test can be more interpretable, making it easier to be deployed in practice to obtain a binary uncertainty decision. To quantify how confident our model is about this prediction, we evaluate whether the difference between the empirical distributions of the two most possible classes from multiple posterior samples is statistically significant. Please see Appendix D for a detailed explanation of the test procedure.

**Uncertainty evaluation via PAvPU:** With the $p$-value of the testing result and a given $p$-value threshold, we can determine whether the model is certain or uncertain about one prediction. To evaluate the uncertainty estimates, we uses Patch Accuracy vs Patch Uncertainty (PAvPU) (Mukhoti & Gal, 2018), which is defined as $\text{PAvPU} = (n_{ac} + n_{iu})/(n_{ac} + n_{au} + n_{ic} + n_{iu})$, where $n_{ac}, n_{au}, n_{ic}, n_{iu}$ are the numbers of accurate and certain, accurate and uncertain, inaccurate and certain, inaccurate and uncertain samples, respectively. This PAvPU evaluation metric would be higher if the model tends to generate the accurate prediction with high certainty and inaccurate prediction with high uncertainty.

### 3.1 CONTEXTUAL DROPOUT ON FULLY CONNECTED LAYERS

We consider an MLP with two hidden layers of size 300 and 100, respectively, with ReLU activations. Dropout is applied to the input layer and the outputs of first two full-connected layers. We use MNIST as the benchmark. We compare contextual dropout with MC dropout (Gal & Ghahramani, 2016), concrete dropout (Gal et al., 2017), Gaussian dropout (Srivastava et al., 2014), and Bayes by Backprop (Blundell et al., 2015). Please see the detailed experimental setting in Appendix C.1.

Table 1: Results on noisy MNIST with MLP.

| METHODS | ACCURACY | PAvPU(0.05) | LOG LIKELIHOOD |
|---|---|---|---|
| MC - BERNOULLI | 86.36 | 85.63 | -1.72 |
| MC - GAUSSIAN | 86.31 | 85.64 | -1.72 |
| CONCRETE | 86.52 | 86.77 | -1.68 |
| BAYES BY BACKPROP | 86.55 | 87.13 | -2.30 |
| CONTEXTUAL GATING | 86.20 | - | -1.81 |
| CONTEXTUAL GATING+DROPOUT | 86.70 | 87.01 | -1.71 |
| BERNOULLI CONTEXTUAL | **87.43**±0.39 | **87.81**±0.23 | **-1.41** ±0.01 |
| GAUSSIAN CONTEXTUAL | 87.35±0.33 | 87.72±0.29 | -1.43±0.01 |

**Results and analysis:** In Table 1, we show accuracy, PAvPU ($p$-value threshold equal to 0.05) and, test predictive loglikelihood with error bars (5 random runs) for models with different dropouts under the challenging noisy data[3] (added Gaussian noise with mean 0, variance 1). Note that the uncertainty

---

[3]Results on original data is deferred to Table 6 in Appendix .

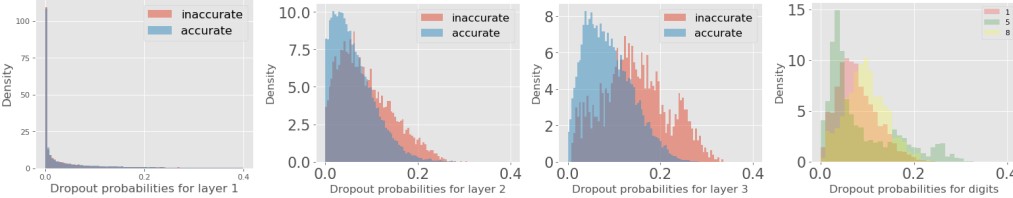

Figure 3: Visualization of dropout probabilities of Bernoulli contextual dropout on the MNIST dataset: the learned dropout probabilities seem to increase as we go to higher-level layers, as also observed in Gal et al. (2017). With contextual dropout, different samples own different dropout probabilities. Inaccurate ones often have higher dropout probabilities corresponding to higher uncertainties. On the further right figure, we compare the dropout distributions across 3 representative digits. The dropout probabilities are overall higher for digit 8 compared to digit 1, meaning 1 is easier to classify. The distribution for 5 has longer tails than others showing there are more variations in the uncertainty for digit 5.

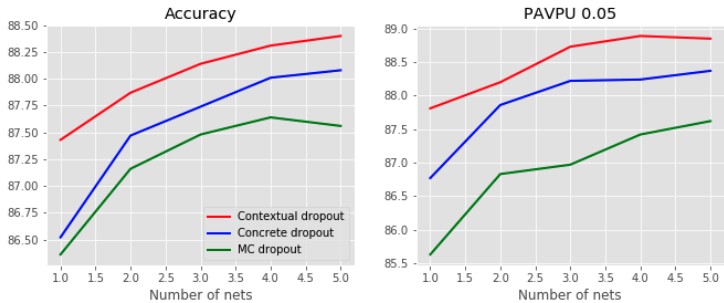

Figure 4: The performance of combining different dropouts with deep ensemble on noisy MNIST data.

results for $p$-value threshold $0.05$ is in general consistent with the results for other $p$-value thresholds (see more in Table 6 in Appendix). We observe that contextual dropout outperforms other methods in all metrics. Moreover, compared to Bayes by Backprop, contextual dropout is more memory and computationally efficient. As shown in Table 5 in Appendix, contextual dropout only introduces $16\%$ additional parameters. However, Bayes by Backprop doubles the memory and increases the computations significantly as we need multiple draws of NN weights for uncertainty. Due to this reason, we do not include it for the following large model evaluations. We note that using the output of the gating network to directly scale activations (contextual gating) underperforms contextual dropout, which shows that the sampling process is important for preventing overfitting and improving robustness to noise. Adding a regular dropout layer on the gating activations (contextual gating + dropout) improves a little, but still underperforms contextual dropout, demonstrating that how we use the gating activations matters. In Figure 3, we observe that Bernoulli contextual dropout learns different dropout probabilities for different samples adapting the sample-level uncertainty which further verifies our motivation and supports the empirical improvements. For sample-dependent dropout, the dropout probabilities would not vanish to zero even though the prior for regularization is also learned, because the optimal dropout probabilities for each sample is not necessarily zero. Enabling different samples to have different network connections could greatly enhance the model's capacity. The prior distribution also plays a different role here. Instead of preventing the dropout probabilities from going to zero, the prior tries to impose some similarities between the dropout probabilities of different samples.

**Combine contextual dropout with Deep Ensemble:** Deep ensemble proposed by Lakshminarayanan et al. (2017) is a simple way to obtain uncertainty by ensembling models trained independently from different random initializations. In Figure 4, we show the performance of combining different dropouts with deep ensemble on noisy MNIST data. As the number of NNs increases, both accuracy and PAvPU increase for all dropouts. However, Bernoulli contextual dropout outperforms other dropouts by a large margin in both metrics, showing contextual dropout is compatible with deep ensemble and their combination can lead to significant improvements. **Out of distribution (OOD) evaluation:** we evaluate different dropouts in an OOD setting, where we train our model with clean data but test it on noisy data. Contextual dropout achieves accuracy of $78.08$, consistently higher than MC dropout ($75.22$) or concrete dropout ($74.93$). Meanwhile, the proposed method is also better at uncertainty estimation with PAvPU of $78.49$, higher than MC ($74.61$) or Concrete ($75.49$).

## 3.2 Contextual dropout on convolutional layers

We apply dropout to the convolutional layers in WRN (Zagoruyko & Komodakis, 2016). In Figure 6 in Appendix, we show the architecture of WRN, where dropout is applied to the first convolutional layer in each network block; in total, dropout is applied to 12 convolutional layers. We evaluate on CIFAR-10 and CIFAR-100 . The detailed setting is provided in Appendix C.1.

Table 2: Results on CIFAR-100 with WRN.

| Dropout | Original Data | | | Noisy Data | | |
|---|---|---|---|---|---|---|
| | Accuracy | PAvPU (0.05) | log likelihood | Accuracy | PAvPU (0.05) | log likelihood |
| Bernoulli | 79.03 | 61.54 | -4.49 | 52.01 | 54.25 | -4.55 |
| Gaussian | 76.63 | 78.05 | -3.93 | 51.38 | 57.02 | -4.23 |
| Concrete | 79.19 | 64.14 | -4.50 | 51.58 | 56.61 | -4.56 |
| Bernoulli Contextual | 80.85±0.05 | 81.56±0.31 | -3.56±0.02 | 53.64±0.45 | **58.63**±0.50 | **-3.73**±0.04 |
| Gaussian Contextual | **80.93**±0.18 | **81.69**±0.16 | **-3.43**±0.07 | **53.72**±0.34 | 58.49±0.43 | -3.81 ±0.03 |

**Results and analysis:** We show the results for CIFAR-100 in Table 2 (see CIFAR-10 results in Tables 8-9 in Appendix). Accuracies, PAvPUs, and test predictive loglikelihoods are incorporated for both the original and noisy data. We consistently observe that contextual dropout outperforms other models in accuracy, uncertainty estimation, and loglikelihood.

**Uncertainty visualization:** We conducted extensive qualitative analyses for uncertainty evaluation. In Figures 9-11 in Appendix F.2, we visualize 15 CIFAR images (with true label) and compare the corresponding probability outputs of different dropouts in boxplots. We observe (1) contextual dropout predicts the correct answer if it is certain, (2) contextual dropout is certain and predicts the correct answers on many images for which MC or concrete dropout is uncertain, (3) MC or concrete dropout is uncertain about some easy examples or certain on some wrong predictions (see details in Appendix F.2), (4) on an image that all three methods have high uncertainty, contextual dropout places a higher probability on the correct answer than the other two. These observations verify that contextual dropout provides better calibrated uncertainty.

Table 3: Results on ImageNet with ResNet-18.

| Dropout | Top-1 Acc | PAvPU | Params | sec/step |
|---|---|---|---|---|
| Without | 69.75 | NA | 11.70M | 1.44 |
| +Gaussian | 69.46 | 72.86 | 11.70M | 1.50 |
| +Contextual | **70.03**±0.07 | **74.68**±0.08 | 11.88M | 1.64 |
| +Contextual (scratch) | **70.29**±0.09 | **76.47**±0.12 | 11.88M | 1.64 |

**Large-scale experiments with ImageNet:** Contextual dropout is also applied to the convolutional layers in ResNet-18, where we plug contextual dropout into a pretrained model, and fine-tune the pretrained model on ImageNet. In Table 3, we show it is even possible to finetune a pretrained model with contextual dropout module, and without much additional memory or run time cost, it achieves better performance than both the original model and the one with regular Gaussian dropout. Training model with contextual dropout from scratch can further improve the performance. See detailed experimental setting in Appendix C.1.

## 3.3 Contextual dropout on attention layers

We further apply contextual dropout to the attention layers of VQA models, whose goal is to provide an answer to a question relevant to the content of a given image. We conduct experiments on the commonly used benchmark, VQA-v2 (Goyal et al., 2017), containing human-annotated question-answer (QA) pairs. There are three types of questions: Yes/No, Number, and Other. In Figure 5, we show one example for each question type. There are 10 answers provided by 10 different human annotators for each question (see explanation of evaluation metrics in Appendix C.2). As shown in the examples, VQA is generally so challenging that there are often several different human annotations for a given image. Therefore, good uncertainty estimation becomes even more necessary.

**Model and training specifications:** We use MCAN (Yu et al., 2019), a state-of-the-art Transformer-like model for VQA. Self-attention layers for question features and visual features, as well as the question-guided attention layers of visual features, are stacked one over another to build a deep model. Dropout is applied in every attention layer (after the softmax and before residual layer (Vaswani et al., 2017)) and fully-connected layer to prevent overfitting (Yu et al., 2019), resulting in 62 dropout layers

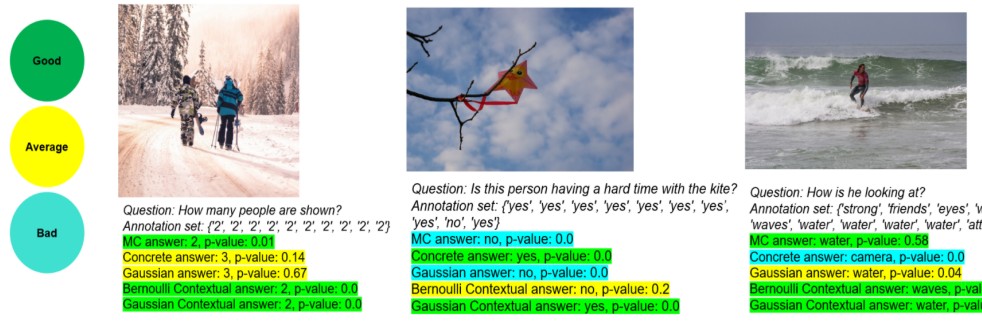

Figure 5: VQA visualization: for each question type, we present an image-question pair along with human annotations. We manually classify each prediction by different methods based on their answers and $p$-values. For questions that have a clear answer, we define the good as certain & accurate, the average as uncertain & accurate or uncertain & inaccurate, and the bad as certain & inaccurate. Otherwise, we define the good as uncertain & accurate, the average as certain & accurate or uncertain & inaccurate, and the bad as certain & inaccurate.

in total. Experiments are conducted using the code of Yu et al. (2019) as basis. Detailed experiment setting is in Appendix C.2.

Table 4: Accuracy and PAvPU on visual question answering.

| Dropout | Accuracy | | PAvPU | |
|---|---|---|---|---|
| | Original Data | Noisy Data | Original Data | Noisy Data |
| Bernoulli (Yu et al., 2019) | 67.2 | - | - | - |
| MC - Bernoulli | 66.95 | 61.45 | 70.04 | 66.11 |
| MC - Gaussian | 66.96 | 62.75 | 70.77 | 67.42 |
| Concrete | 66.82 | 61.47 | 71.02 | 65.94 |
| Bernoulli Contextual | **67.42**±0.06 | 63.73±0.08 | **71.65**±0.06 | 68.57±0.11 |
| Gaussian Contextual | 67.35±0.03 | **63.82**±0.05 | 71.62±0.02 | **68.64**±0.04 |

**Results and analysis:** We compare different dropouts on both the original VQA dataset and a noisy version, where Gaussian noise with standard deviation 5 is added to the visual features. In Tables 4, we show the overall accuracy and uncertainty estimation. The results show that on the original data, contextual dropout achieves better accuracy and uncertainty estimation than the others. Moreover, on noisy data, where the prediction becomes more challenging and requires more model flexibility and robustness, contextual dropouts outperform their regular dropout counterparts by a large margin in terms of accuracy with consistent improvement across all three question types.

**Visualization:** In Figures 12-15 in Appendix F.3, we visualize some image-question pairs, along with the human annotations and compare the predictions and uncertainty estimations of different dropouts. We show three of them in Figure 5. As shown in the plots, overall contextual dropout is more conservative on its wrong predictions and more certain on its correct predictions than other methods (see more detailed explanations in Appendix F.3).

## 4 Conclusion

We introduce contextual dropout as a simple and scalable data-dependent dropout module that achieves strong performance in both accuracy and uncertainty estimation on a variety of tasks including large scale applications. With an efficient parameterization of the coviariate-dependent variational distribution, contextual dropout boosts the flexibility of Bayesian neural networks with only slightly increased memory and computational cost. We demonstrate the general applicability of contextual dropout on fully connected, convolutional, and attention layers, and also show that contextual dropout masks are compatible with both Bernoulli and Gaussian distribution. On both image classification and visual question answering tasks, contextual dropout consistently outperforms corresponding baselines. Notably, on ImageNet, we find it is possible to improve the performance of a pretrained model by adding the contextual dropout module during a finetuning stage. Based on these results, we believe contextual dropout can serve as an efficient alternative to data-independent dropouts in the versatile tool box of dropout modules.

ACKNOWLEDGEMENTS

The authors acknowledge the support of Grants IIS-1812699, IIS-1812641, ECCS-1952193, CCF-1553281, and CCF-1934904 from the U.S. National Science Foundation, and the Texas Advanced Computing Center for providing HPC resources that have contributed to the research results reported within this paper. M. Zhou acknowledges the support of a gift fund from ByteDance Inc.

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

# Appendix

## A  DETAILS OF ARM GRADIENT ESTIMATOR FOR BERNOULLI CONTEXTUAL DROPOUT

In this section, we will explain the implementation details of ARM for Bernoulli contextual dropout. To compute the gradients with respect to the parameters of the variational distribution, a commonly used gradient estimator is the REINFORCE estimator (Williams, 1992) as

$$\nabla_{\boldsymbol{\varphi}}\mathcal{L}(\boldsymbol{x}, y) = \mathbb{E}_{\boldsymbol{z}\sim q_{\boldsymbol{\phi}}(\cdot\,|\,\boldsymbol{x})}[r(\boldsymbol{x}, \boldsymbol{z}, y)\nabla_{\boldsymbol{\varphi}}\log q_{\boldsymbol{\phi}}(\boldsymbol{z}\,|\,\boldsymbol{x})], \quad r(\boldsymbol{x}, \boldsymbol{z}, y) := \log\frac{p_{\boldsymbol{\theta}}(y\,|\,\boldsymbol{x}, \boldsymbol{z})p_{\boldsymbol{\eta}}(\boldsymbol{z})}{q_{\boldsymbol{\phi}}(\boldsymbol{z}\,|\,\boldsymbol{x})}.$$

This gradient estimator is, however, known to have high variance (Yin & Zhou, 2018). To mitigate this issue, we use ARM to compute the gradient with Bernoulli random variable.

**ARM gradient estimator:**  In general, denoting $\sigma(\boldsymbol{\alpha}) = 1/(1 + e^{-\boldsymbol{\alpha}})$ as the sigmoid function, ARM expresses the gradient of $\mathcal{E}(\boldsymbol{\alpha}) = \mathbb{E}_{\boldsymbol{z}\sim\prod_{k=1}^{K}\mathrm{Ber}(z_k;\sigma(\boldsymbol{\alpha}_k))}[r(\boldsymbol{z})]$ as

$$\nabla_{\boldsymbol{\alpha}}\mathcal{E}(\boldsymbol{\alpha}) = \mathbb{E}_{\boldsymbol{\pi}\sim\prod_{k=1}^{K}\mathrm{Uniform}(\pi_k;0,1)}[g_{\mathrm{ARM}}(\boldsymbol{\pi})], \quad g_{\mathrm{ARM}}(\boldsymbol{\pi}) := [r(\boldsymbol{z}_{\mathrm{true}}) - r(\boldsymbol{z}_{\mathrm{sudo}})](1/2 - \boldsymbol{\pi}), \quad (6)$$

where $\boldsymbol{z}_{\mathrm{true}} := \mathbf{1}_{[\boldsymbol{\pi}<\sigma(\boldsymbol{\alpha})]}$ and $\boldsymbol{z}_{\mathrm{sudo}} := \mathbf{1}_{[\boldsymbol{\pi}>\sigma(-\boldsymbol{\alpha})]}$ are referred to as the true and pseudo actions, respectively, and $\mathbf{1}_{[\cdot]} \in \{0, 1\}^K$ is an indicator function.

**Sequential ARM:**  Note that the above equation is not directly applicable to our model due to the cross-layer dependence. However, the dropout masks within each layer are independent of each other conditioned on these of the previous layers, so we can break our expectation into a sequence and apply ARM sequentially. We rewrite $\mathcal{L} = \mathbb{E}_{\boldsymbol{z}\sim q_{\boldsymbol{\phi}}(\cdot\,|\,\boldsymbol{x})}[r(\boldsymbol{x}, \boldsymbol{z}, y)]$. When computing $\nabla_{\boldsymbol{\varphi}}\mathcal{L}$, we can ignore the $\boldsymbol{\varphi}$ in $r$ as the expectation of $\nabla_{\boldsymbol{\varphi}}\log q_{\boldsymbol{\phi}}(\boldsymbol{z}\,|\,\boldsymbol{x})$ is zero. Using the chain rule, we have $\nabla_{\boldsymbol{\varphi}}\mathcal{L} = \sum_{l=1}^{L}\nabla_{\boldsymbol{\alpha}^l}\mathcal{L}\nabla_{\boldsymbol{\varphi}}\boldsymbol{\alpha}^l$. With decomposition $\mathcal{L} = \mathbb{E}_{\boldsymbol{z}^{1:l-1}\sim q_{\boldsymbol{\phi}}(\cdot\,|\,\boldsymbol{x})}\mathbb{E}_{\boldsymbol{z}^l\sim\mathrm{Ber}(\sigma(\boldsymbol{\alpha}^l))}[r(\boldsymbol{x}, \boldsymbol{z}^{1:l}, y)]$, where $r(\boldsymbol{x}, \boldsymbol{z}^{1:l}, y) := \mathbb{E}_{\boldsymbol{z}^{l+1:L}\sim q_{\boldsymbol{\phi}}(\cdot\,|\,\boldsymbol{x}, \boldsymbol{z}^{1:l})}[r(\boldsymbol{x}, \boldsymbol{z}, y)]$, we know

$$\nabla_{\boldsymbol{\alpha}^l}\mathcal{L} = \mathbb{E}_{\boldsymbol{z}^{1:l-1}\sim q_{\boldsymbol{\phi}}(\cdot\,|\,\boldsymbol{x})}\mathbb{E}_{\boldsymbol{\pi}^l\sim\prod_k\mathrm{Uniform}(\pi_k^l;0,1)}[g_{\mathrm{ARM}}(\boldsymbol{\pi}^l)],$$

$$g_{\mathrm{ARM}}(\boldsymbol{\pi}^l) = [r(\boldsymbol{x}, \boldsymbol{z}^{1:l-1}, \boldsymbol{z}_{\mathrm{true}}^l, y) - r(\boldsymbol{x}, \boldsymbol{z}^{1:l-1}, \boldsymbol{z}_{\mathrm{sudo}}^l, y)](1/2 - \boldsymbol{\pi}^l),$$

where $\boldsymbol{z}_{\mathrm{true}}^l := \mathbf{1}_{[\boldsymbol{\pi}^l<\sigma(\boldsymbol{\alpha}^l)]}$ and $\boldsymbol{z}_{\mathrm{sudo}}^l := \mathbf{1}_{[\boldsymbol{\pi}^l>\sigma(-\boldsymbol{\alpha}^l)]}$. We estimate the gradients via Monte Carlo integration. We provide the pseudo code in Algorithm 1.

**Implementation details:**  The computational complexity of sequential ARM is $O(L)$ times of that of the decoder computation. Although it is embarrassingly parallelizable, in practice, with limited computational resource available, it maybe be challenging to use sequential ARM when $L$ is fairly large. In such cases, the original non-sequential ARM can be viewed as an approximation to strike a good balance between efficiency and accuracy (see the pseudo code in Algorithm 2 in Appendix). In our cases, for image classification models, $L$ is small enough (3 for MLP, 12 for WRN) for us to use sequential ARM. For VQA, $L$ is as large as 62 and hence we choose the non-sequential ARM.

To control the learning rate of the encoder, we use a scaled sigmoid function: $\sigma_t(\boldsymbol{\alpha}^l) = \frac{1}{1+\exp(-t\boldsymbol{\alpha}^l)}$, where a larger $t$ corresponding to a larger learning rate for the encoder. This function is also used in Li & Ji (2019) to facilitate the transition of probability between 0 and 1 for the purpose of pruning NN weights.

# B    ALGORITHMS

Below, we present training algorithms for both Bernoulli and Gaussian contextual dropout.

---

**Algorithm 1:** Bernoulli contextual dropout with sequential ARM

---

**Input:** data $\mathcal{D}$, $r$, $\{g_{\boldsymbol{\theta}}^l\}_{l=1}^L$, $\{h_{\boldsymbol{\varphi}}^l\}_{l=1}^L$, step size $s$
**Output:** updated $\boldsymbol{\theta}$, $\boldsymbol{\varphi}$, $\boldsymbol{\eta}$
**repeat**
  $G_{\boldsymbol{\varphi}} = 0$;
  Sample $\boldsymbol{x}, y$ from data $\mathcal{D}$;
  $\boldsymbol{x}^0 = \boldsymbol{x}$
  **for** $l = 1$ **to** $L$ **do**
    $U^l = g_{\boldsymbol{\theta}}^l(\boldsymbol{x}^{l-1})$, $\boldsymbol{\alpha}^l = h_{\boldsymbol{\varphi}}^l(U^l)$
    Sample $\boldsymbol{\pi}^l$ from Uniform(0,1);
    $\boldsymbol{z}_{\text{true}}^l := \mathbf{1}_{[\boldsymbol{\pi}^l < \sigma_t(\boldsymbol{\alpha}^l)]}$;
    $\boldsymbol{z}_{\text{sudo}}^l := \mathbf{1}_{[\boldsymbol{\pi}^l > \sigma_t(-\boldsymbol{\alpha}^l)]}$;
    **if** $\boldsymbol{z}_{\text{true}}^l = \boldsymbol{z}_{\text{sudo}}^l$ **then**
      $r_{\text{sudo}}^l =$None;
    **else**
      $\boldsymbol{x}_{\text{sudo}}^l = U^l \odot \boldsymbol{z}_{l,\text{sudo}}$
      **for** $k = l + 1$ **to** $L$ **do**
        $U_{\text{sudo}}^k = g_{\boldsymbol{\theta}}^k(\boldsymbol{x}_{\text{sudo}}^{k-1})$, $\boldsymbol{\alpha}_{\text{sudo}}^k = h_{\boldsymbol{\varphi}}^k(U_{\text{sudo}}^k)$
        Sample $\boldsymbol{\pi}_{\text{sudo}}^k$ from Uniform(0,1);
        $\boldsymbol{z}_{\text{sudo}}^k := \mathbf{1}_{[\boldsymbol{\pi}_{\text{sudo}}^k < \sigma_t(\boldsymbol{\alpha}_{\text{sudo}}^k)]}$;
        $\boldsymbol{x}_{\text{sudo}}^k = U_{\text{sudo}}^k \odot \boldsymbol{z}_{k,\text{sudo}}$;
      **end for**
      $r_{\text{sudo}}^l = r(\boldsymbol{x}_{\text{sudo}}^L, y)$
    **end if**
    $\boldsymbol{x}^l = U^l \odot \boldsymbol{z}_{\text{true}}^l$
  **end for**
  $r_{\text{true}} = r(\boldsymbol{x}_{\text{true}}^L, y)$
  **for** $l = 1$ **to** $L$ **do**
    **if** $r_{\text{sudo}}^l$ is not None **then**
      $G_{\boldsymbol{\varphi}} = G_{\boldsymbol{\varphi}} + t(r_{\text{true}} - r_{\text{sudo}}^l)(1/2 - \boldsymbol{\pi}^l)\nabla_{\boldsymbol{\varphi}}\boldsymbol{\alpha}^l$ ;
    **end if**
  **end for**
  $\boldsymbol{\varphi} = \boldsymbol{\varphi} + sG_{\boldsymbol{\varphi}}$, with step-size $s$;
  $\boldsymbol{\theta} = \boldsymbol{\theta} + s\frac{\partial \log p_{\boldsymbol{\theta}}(y \mid \boldsymbol{x}, \boldsymbol{z}_{1:L,\text{true}})}{\partial \boldsymbol{\theta}}$;
  $\boldsymbol{\eta} = \boldsymbol{\eta} + s\frac{\partial \log p_{\boldsymbol{\eta}}(\boldsymbol{z}_{1:L,\text{true}})}{\partial \boldsymbol{\eta}}$;
**until** convergence

---

---

**Algorithm 2:** Bernoulli contextual dropout with independent ARM

---

**Input:** data $\mathcal{D}$, $r$, $\{g_{\boldsymbol{\theta}}^l\}_{l=1}^L$, $\{h_{\boldsymbol{\varphi}}^l\}_{l=1}^L$, step size $s$
**Output:** updated $\boldsymbol{\theta}$, $\boldsymbol{\varphi}$, $\boldsymbol{\eta}$
**repeat**
    $G_{\boldsymbol{\varphi}} = 0$;
    Sample $\boldsymbol{x}, y$ from data $\mathcal{D}$;
    $\boldsymbol{x}^0 = \boldsymbol{x}$
    **for** $l = 1$ **to** $L$ **do**
        $U^l = g_{\boldsymbol{\theta}}^l(\boldsymbol{x}^{l-1})$, $\boldsymbol{\alpha}^l = h_{\boldsymbol{\varphi}}^l(U^l)$
        Sample $\boldsymbol{\pi}^l$ from Uniform(0,1);
        $\boldsymbol{z}_{\text{true}}^l := \mathbf{1}_{[\boldsymbol{\pi}^l < \sigma_t(\boldsymbol{\alpha}^l)]}$;
        $\boldsymbol{x}^l = U^l \odot \boldsymbol{z}_{\text{true}}^l$
    **end for**
    $r_{\text{true}} = r(\boldsymbol{x}_{\text{true}}^L, y)$
    $\boldsymbol{x}_{\text{sudo}}^0 = \boldsymbol{x}$
    **for** $l = 1$ **to** $L$ **do**
        $U_{\text{sudo}}^l = g_{\boldsymbol{\theta}}^l(\boldsymbol{x}_{\text{sudo}}^{l-1})$, $\boldsymbol{\alpha}_{\text{sudo}}^l = h_{\boldsymbol{\varphi}}^l(U_{\text{sudo}}^l)$
        $\boldsymbol{z}_{\text{sudo}}^l := \mathbf{1}_{[\boldsymbol{\pi}_{\text{sudo}}^l > \sigma_t(-\boldsymbol{\alpha}_{\text{sudo}}^l)]}$;
        $\boldsymbol{x}_{\text{sudo}}^l = U_{\text{sudo}}^l \odot \boldsymbol{z}_{\text{sudo}}^l$
    **end for**
    $r_{\text{sudo}} = r(\boldsymbol{x}_{\text{sudo}}^L, y)$;
    **for** $l = 1$ **to** $L$ **do**
        $G_{\boldsymbol{\varphi}} = G_{\boldsymbol{\varphi}} + t(r_{\text{true}} - r_{\text{sudo}})(1/2 - \boldsymbol{\pi}^l)\nabla_{\boldsymbol{\varphi}}\boldsymbol{\alpha}^l$ ;
    **end for**
    $\boldsymbol{\varphi} = \boldsymbol{\varphi} + sG_{\boldsymbol{\varphi}}$, with step-size $s$;
    $\boldsymbol{\theta} = \boldsymbol{\theta} + s\frac{\partial \log p_{\boldsymbol{\theta}}(y \mid \boldsymbol{x}, \boldsymbol{z}_{1:L,\text{true}})}{\partial \boldsymbol{\theta}}$;
    $\boldsymbol{\eta} = \boldsymbol{\eta} + s\frac{\partial \log p_{\boldsymbol{\eta}}(\boldsymbol{z}_{1:L,\text{true}})}{\partial \boldsymbol{\eta}}$;
**until** convergence

---

**Algorithm 3:** Gaussian contextual dropout with reparamaterization trick

**Input:** data $\mathcal{D}$, $r$, $\{g_{\boldsymbol{\theta}}^l\}_{l=1}^L$, $\{h_{\boldsymbol{\varphi}}^l\}_{l=1}^L$, step size $s$
**Output:** updated $\boldsymbol{\theta}$, $\boldsymbol{\varphi}$, $\boldsymbol{\eta}$
**repeat**
    Sample $\boldsymbol{x}, y$ from data $\mathcal{D}$;
    $\boldsymbol{x}^0 = \boldsymbol{x}$
    **for** $l = 1$ **to** $L$ **do**
        $U^l = g_{\boldsymbol{\theta}}^l(\boldsymbol{x}^{l-1})$, $\boldsymbol{\alpha}^l = h_{\boldsymbol{\varphi}}^l(U^l)$
        Sample $\boldsymbol{\epsilon}^l$ from $\mathcal{N}(0, 1)$;
        $\boldsymbol{\tau}^l = \sqrt{\frac{1 - \sigma_t(\boldsymbol{\alpha}^l)}{\sigma_t(\boldsymbol{\alpha}^l)}}$;
        $\boldsymbol{z}^l := \boldsymbol{1} + \boldsymbol{\tau}^l \odot \boldsymbol{\epsilon}^l$;
        $\boldsymbol{x}^l = U^l \odot \boldsymbol{z}^l$
    **end for**
    $\boldsymbol{\varphi} = \boldsymbol{\varphi} + s \nabla_{\boldsymbol{\varphi}} (\log p_{\boldsymbol{\theta}}(y \,|\, \boldsymbol{x}, \boldsymbol{z}_{1:L}) - \frac{\log q_{\boldsymbol{\phi}}(\boldsymbol{z}_{1:L}|\boldsymbol{x})}{\log p_{\boldsymbol{\eta}}(\boldsymbol{z}_{1:L})})$, with step-size $s$;
    $\boldsymbol{\theta} = \boldsymbol{\theta} + s \frac{\partial \log p_{\boldsymbol{\theta}}(y \,|\, \boldsymbol{x}, \boldsymbol{z}_{1:L})}{\partial \boldsymbol{\theta}}$;
    $\boldsymbol{\eta} = \boldsymbol{\eta} + s \frac{\partial \log p_{\boldsymbol{\eta}}(\boldsymbol{z}_{1:L})}{\partial \boldsymbol{\eta}}$;
**until** convergence

## C    Details of Experiments

All experiments are conducted using a single Nvidia Tesla V100 GPU.

Table 5: Model size comparison among different methods.

| Method | MLP | WRN | MCAN | ResNet-18 |
|---|---|---|---|---|
| MC or Concrete | 267K | 36.5M | 58M | 11.6M |
| Contextual | 311K | 36.6M | 61M | 11.8M |
| Bayes By Backprop | 534K | - | - | - |

**Choice of hyper-parameters in Contextual Dropout:** Contextual dropout introduces two additional hyperparameters compared to regular dropout. One is the channel factor $\gamma$ for the encoder network. In our experiments, the results are not sensitive to the choice of the value of the channel factor $\gamma$. Any number from 8 to 16 would give similar results, which is also observed in (Hu et al., 2018). The other is the sigmoid scaling factor $t$ that controls the learning rate of the encoder. We find that the performance is not that sensitive to its value and it is often beneficial to make it smaller than the learning rate of the decoder. In all experiments considered in the paper, which cover various noise levels and model sizes, we have simply fixed it at $t = 0.01$.

### C.1    Image Classification

**MLP:** We consider an MLP with two hidden layers of size 300 and 100, respectively, and use ReLU activations. Dropout is applied to all three full-connected layers. We use MNIST as the benchmark. All models are trained for 200 epochs with batch size 128 and the Adam optimizer (Kingma & Ba, 2014) ($\beta_1 = 0.9$, $\beta_2 = 0.999$). The learning rate is 0.001. We compare contextual dropout with MC dropout (Gal & Ghahramani, 2016) and concrete dropout (Gal et al., 2017). For MC dropout, we use the hand-tuned dropout rate at 0.2. For concrete dropout, we initialize the dropout rate at 0.2 for Bernoulli dropout and the standard deviation parameter at 0.5 for Gaussian dropout. and set the Concrete temperature at 0.1 (Gal et al., 2017). We initialize the weights in contextual dropout with *He-initialization* preserving the magnitude of the variance of the weights in the forward pass (He et al., 2015). We initialize the biases in the way that the dropout rate is 0.2 when the weights for contextual dropout are zeros. We also initialize our prior dropout rate at 0.2. For hyperparameter tuning, we hold out $10,000$ samples randomly selected from the training set for validation. We use the chosen hyperparameters to train on the full training set ($60,000$ samples) and evaluate on the testing set ($10,000$ samples). We use Leaky ReLU (Xu et al., 2015a) with 0.1 as the non-linear

operator in contextual dropout. The reduction ratio $\gamma$ is set as 10, and sigmoid scaling factor $t$ as $0.01$. For Bayes by Backprop, we use $-\log \sigma_1 = 0, -\log \sigma_2 = 6, \pi = 0.2$ (following the notation in the original paper). For evaluation, we set $M = 20$.

**WRN:** We consider WRN (Zagoruyko & Komodakis, 2016), including 25 convolutional layers. In Figure 6, we show the architecture of WRN, where dropout is applied to the first convolutional layer in each network block; in total, dropout is applied to 12 convolutional layers. We use CIFAR-10 and CIFAR-100 (Krizhevsky et al., 2009) as benchmarks. All experiments are trained for 200 epochs with the Nesterov Momentum optimizer (Nesterov, 1983), whose base learning rate is set as $0.1$, with decay factor $1/5$ at epochs 60 and 120. All other hyperparameters are the same as MLP except for Gaussian dropout, where we use standard deviation equal to 0.8 for the CIFAR100 with no noise and 1 for all other cases.

**ResNet:** We used ResNet-18 as the baseline model. We use momentum SGD, with learning rate $0.1$, and momentum weight $0.9$. Weight decay is utilized with weight $1e^{-4}$. For models trained from scratch, we train the models with 90 epochs. For finetuning models, we start with pretrained baseline ResNet models and finetune for 1 epoch.

| Group Name | Layers |
|---|---|
| conv1 | [Original Conv (16)] |
| conv2 | [Conv + Dropout (160); Original Conv (160)] x 4 |
| conv3 | [Conv + Dropout (320); Original Conv (320)] x 4 |
| conv4 | [Conv + Dropout (640); Original Conv (640)] x 4 |

Figure 6: Architecture of the Wide Residual Network.

## C.2 VQA

**Dataset**: The dataset is split into the training (80k images and 444k QA pairs), validation (40k images and 214k QA pairs), and testing (80k images and 448k QA pairs) sets. We perform evaluation on the validation set as the true labels for the test set are not publicly available (Deng et al., 2018).

**Evaluation metric:** the evaluation for VQA is different from image classification. The accuracy for a single answer could be a number between 0 and 1 (Goyal et al., 2017): $\text{Acc}(ans) = \min\{(\#\text{human that said } ans)/3, 1\}$. We generalize the uncertainty evaluation accordingly:

$$n_{ac} = \sum_i \text{Acc}_i \text{Cer}_i, \; n_{iu} = \sum_i (1 - \text{Acc}_i)(1 - \text{Cer}_i), \; n_{au} = \sum_i \text{Acc}_i (1 - \text{Cer}_i), \; n_{ic} = \sum_i (1 - \text{Acc}_i)(\text{Cer}_i)$$

where for the $i$th prediction $\text{Acc}_i$ is the accuracy and $\text{Cer}_i \in \{0, 1\}$ is the certainty indicator.

**Experimental setting:** We follow the setting by Yu et al. (2019), where bottom-up features extracted from images by Faster R-CNN (Ren et al., 2015) are used as visual features, pretrained word-embeddings (Pennington et al., 2014) and LSTM (Hochreiter & Schmidhuber, 1997) are used to extract question features. We adopt the encoder-decoder structure in MCAN with six co-attention layers. We use the same model hyperparameters and training settings in Yu et al. (2019) as follows: the dimensionality of input image features, input question features, and fused multi-modal features are set to be 2048, 512, and 1024, respectively. The latent dimensionality in the multi-head attention is 512, the number of heads is set to 8, and the latent dimensionality for each head is 64. The size of the answer vocabulary is set to $N = 3129$ using the strategy in Teney et al. (2018). To train the MCAN model, we use the Adam optimizer (Kingma & Ba, 2014) with $\beta_1 = 0.9$ and $\beta_2 = 0.98$. The base learning rate is set to $\min(2.5te^{-5}, 1e^{-4})$, where $t$ is the current epoch number starting from 1. After 10 epochs, the learning rate is decayed by $1/5$ every 2 epochs. All the models are trained up to 13 epochs with the same batch size of 64.

We only conduct training on the training set (no data augmentation with visual genome dataset), and evaluation on the validation set. For MC dropout, we use the dropout rate of $0.1$ for Bernoulli dropout as in Yu et al. (2019) and the standard deviation parameter of $1/3$ for Gaussian dropout. For concrete dropout, we initialize the dropout rate at $0.1$ and set the Concrete temperature at $0.1$ (Gal et al., 2017). For hyperparameter tuning, we randomly hold out $20\%$ of the training set for validation. After tuning, we train on the whole training set and evaluate on the validation set. We initialize the weights with *He-initialization* preserving the magnitude of the variance of the weights in the forward pass (He

et al., 2015). We initialize the biases in the way that the dropout rate is $0.1$ when the weights for contextual dropout are zeros. We also initialize our prior dropout rate at $0.1$. We use ReLU as the non-linear operator in contextual dropout. We use $\gamma = 8$ for layers with $C_d^l > 8$, otherwise $\gamma = 1$. We set $\alpha \in \mathbb{R}^{d_V}$ for residual layers.

## D  STATISTICAL TEST FOR UNCERTAINTY ESTIMATION

Consider $M$ posterior samples of predictive probabilities $\{\boldsymbol{p}_m\}_{m=1}^M$, where $\boldsymbol{p}_m$ is a vector with the same dimension as the number of classes. For single-label classification models, $\boldsymbol{p}_m$ is produced by a softmax layer and sums to one, while for multi-label classification models, $\boldsymbol{p}_m$ is produced by a sigmoid layer and each element is between $0$ and $1$. The former output is used in most image classification models, while the latter is often used in VQA where multiple answers could be true for a single input. In both cases, to quantify how confident our model is about this prediction, we evaluate whether the difference between the probabilities of the first and second highest classes is statistically significant with a statistical test. We conduct the normality test on the output probabilities for both image classification and VQA models, and find most of the output probabilities are approximately normal (we randomly pick some Q-Q plots (Ghasemi & Zahediasl, 2012) and show them in Figures 7 and 8). This motivates us to use two-sample t-test[4]. In the following, we briefly summarize the two-sample $t$-test we use.

Two sample hypothesis testing is an inferential statistical test that determines whether there is a statistically significant difference between the means in two groups. The null hypothesis for the $t$-test is that the population means from the two groups are equal: $\mu_1 = \mu_2$, and the alternative hypothesis is $\mu_1 \neq \mu_2$. Depending on whether each sample in one group can be paired with another sample in the other group, we have either paired $t$-test or independent $t$-test. In our experiments, we utilize both types of two sample $t$-test. For a single-label model, the probabilities are dependent between two classes due to the softmax layer, therefore, we use the *paired* two-sample $t$-test; for a multi-label model, the probabilities are independent given the logits of the output layer, so we use the *independent* two-sample $t$-test.

For paired two-sample $t$-test, we calculate the difference between the paired observations calculate the $t$-statistic as below:

$$T = \frac{\bar{Y}}{s/\sqrt{N}},$$

where $\bar{Y}$ is the mean difference between the paired observations, $s$ is the standard deviation of the differences, and $N$ is the number of observations. Under the null hypothesis, this statistic follows a $t$-distribution with $N - 1$ degrees of freedom if the difference is normally distributed. Then, we use this $t$-statistic and $t$-distribution to calculate the corresponding $p$-value.

For independent two-sample $t$-test, we calculate the $t$-statistic as below:

$$T = \frac{\bar{Y}_1 - \bar{Y}_2}{\sqrt{s^2/N_1 + s^2/N_2}}$$

$$s^2 = \frac{\sum(y_1 - \bar{Y}_1) + \sum(y_2 - \bar{Y}_2)}{N_1 + N_2 - 2}$$

where $N_1$ and $N_2$ are the sample sizes, and $\bar{Y}_1$ and $\bar{Y}_2$ are the sample means. Under the null hypothesis, this statistic follows a $t$-distribution with $N_1 + N_2 - 2$ degrees of freedom if both $y_1$ and $y_2$ are normally distributed. We calculate the $p$-value accordingly.

To justify the assumption of the two-sample $t$-test, we run the normality test on the output probabilities for both image classification and VQA models. We find most of the output probabilities are approximately normal. We randomly pick some Q-Q plots (Ghasemi & Zahediasl, 2012) and show them in Figures 7 and 8.

Table 6: Complete results on MNIST with MLP

| | ORIGINAL DATA | | NOISY DATA | |
|---|---|---|---|---|
| | ACCURACY | PAvPU(0.01 / 0.05 / 0.1) | ACCURACY | PAvPU(0.01 / 0.05 / 0.1) |
| MC DROPOUT - BERNOULLI | 98.62 | 98.25 / 98.39 / 98.44 | 86.36 | 84.29/ 85.63 / 86.10 |
| MC DROPOUT - GAUSSIAN | 98.67 | 98.23 / 98.41/ 98.46 | 86.31 | 83.99 / 85.64 / 86.03 |
| CONCRETE DROPOUT | 98.61 | 98.43/ 98.50 / 98.57 | 86.52 | 85.98 / 86.77/ 86.92 |
| BAYES BY BACKPROP | 98.44 | 98.26 / 98.42 / 98.56 | 86.55 | 86.89/ 87.13/ 87.26 |
| BERNOULLI CONTEXTUAL DROPOUT | **99.08**(0.04) | **98.74**(0.17) / **98.92**(0.08) / **99.09**(0.08) | **87.43**(0.39) | **87.75**(0.24) / **87.81**(0.23) / **87.89**(0.25) |
| GAUSSIAN CONTEXTUAL DROPOUT | 98.92(0.09) | 98.71(0.02) / 98.90(0.08) / 99.03(0.07) | 87.35(0.33) | 87.64(0.19) / 87.72(0.29) / 87.78(0.32) |

Table 7: Loglikelihood on original MNIST with MLP.

| | LOG LIKELIHOOD |
|---|---|
| MC - BERNOULLI | $-1.4840 \pm 0.0004$ |
| MC - GAUSSIAN | $-1.4820 \pm 0.0003$ |
| CONCRETE | $-1.4822 \pm 0.0012$ |
| BAYES BY BACKPROP | $-1.4806 \pm 0.0007$ |
| BERNOULLI CONTEXTUAL | $\mathbf{-1.4537} \pm 0.0005$ |
| GAUSSIAN CONTEXTUAL | $-1.4589 \pm 0.0005$ |

Table 8: Complete results on CIFAR-10 with WRN

| | ORIGINAL DATA | | NOISY DATA | |
|---|---|---|---|---|
| | ACCURACY | PAvPU(0.01 / 0.05 / 0.1) | ACCURACY | PAvPU(0.01 / 0.05 / 0.1) |
| MC DROPOUT - BERNOULLI | 94.58 | 78.73 / 82.34 / 84.21 | 79.51 | 72.89 / 74.43 / 75.04 |
| MC DROPOUT - GAUSSIAN | 93.81 | 92.59 / 93.24 / 93.85 | 79.33 | 80.43 / 81.24 / 82.31 |
| CONCRETE DROPOUT | 94.60 | 73.51 / 78.41 / 81.01 | 79.34 | 72.72 / 73.89 / 74.72 |
| BERNOULLI CONTEXTUAL DROPOUT | 95.92(0.10) | 95.25(0.23) / 95.74(0.12) / 96.02(0.16) | 81.49(0.19) | **82.56**(0.50) / **83.28**(0.31) / **83.91**(0.28) |
| GAUSSIAN CONTEXTUAL DROPOUT | **96.04**(0.1) | **95.42**(0.07) / **95.85**(0.07) / **96.10**(0.06) | **81.64**(0.31) | 82.38 (0.41) / 82.80(0.36) / 83.43(0.36) |

Table 9: Complete log likelihood results on CIFAR-10 with WRN

| | CIFAR-10 | |
|---|---|---|
| | ORIGINAL DATA | NOISY DATA |
| MC DROPOUT - BERNOULLI | -1.91 | -1.93 |
| MC DROPOUT - GAUSSIAN | -1.54 | -1.72 |
| CONCRETE DROPOUT | -1.98 | -2.0 |
| BERNOULLI CONTEXTUAL DROPOUT | -1.24 | **-1.47** |
| GAUSSIAN CONTEXTUAL DROPOUT | **-1.19** | -1.51 |

Table 10: Complete results on CIFAR-100 with WRN

| | ORIGINAL DATA | | NOISY DATA | |
|---|---|---|---|---|
| | ACCURACY | PAvPU(0.01 / 0.05 / 0.1) | ACCURACY | PAvPU(0.01 / 0.05 / 0.1) |
| MC DROPOUT - BERNOULLI | 79.03 | 56.90 / 61.54 / 64.14 | 52.01 | 53.86 / 54.25 / 54.63 |
| MC DROPOUT - GAUSSIAN | 76.63 | 77.35 / 78.05 / 78.26 | 51.38 | 56.83 / 57.02 / 57.31 |
| CONCRETE DROPOUT | 79.19 | 59.45 / 64.14/ 66.63 | 51.58 | 57.62 / 56.61/ 55.89 |
| BERNOULLI CONTEXTUAL DROPOUT | 80.85(0.05) | 81.04(0.28) / 81.56(0.31) / 81.86(0.21) | 53.64(0.45) | **58.29**(0.30) / **58.63**(0.50) / **59.36**(0.49) |
| GAUSSIAN CONTEXTUAL DROPOUT | **80.93** (0.18) | **81.43**(0.1) / **81.69**(0.16) / **82.02**(0.14) | **53.72**(0.34) | 58.01(0.6) / 58.49(0.43) / 58.95(0.37) |

# E    TABLES AND FIGURES FOR $p$-VALUE 0.01, 0.05 AND 0.1

# F    QUALITATIVE ANALYSIS

In this section, we include the Q-Q plots of the output probabilities as the normality test for the assumptions of two-sample $t$-test. In Figure 7, we test the normality of differences between highest probabilities and second highest probabilities on WRN model with contextual dropout trained on the orignal CIFAR-10 dataset. In Figure 8, we test the normality of highest probabilities and second highest probabilities (separately) on VQA model with contextual dropout trained on the original VQA-v2 dataset. We use 20 data points for the plots.

## F.1    NORMALITY TEST OF OUTPUT PROBABILITIES

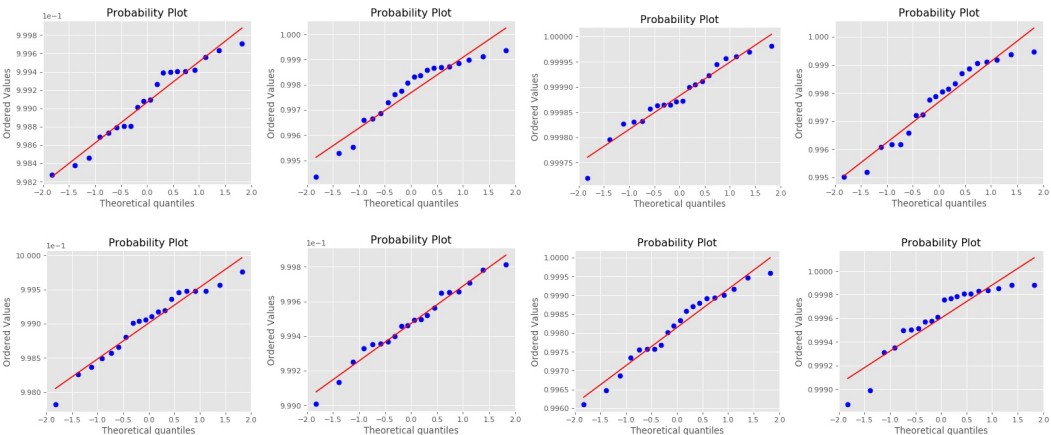

Figure 7: QQ Plot for differences between highest probabilities and second highest probabilities on WRN model with contextual dropout trained on the orignal CIFAR-10 dataset.

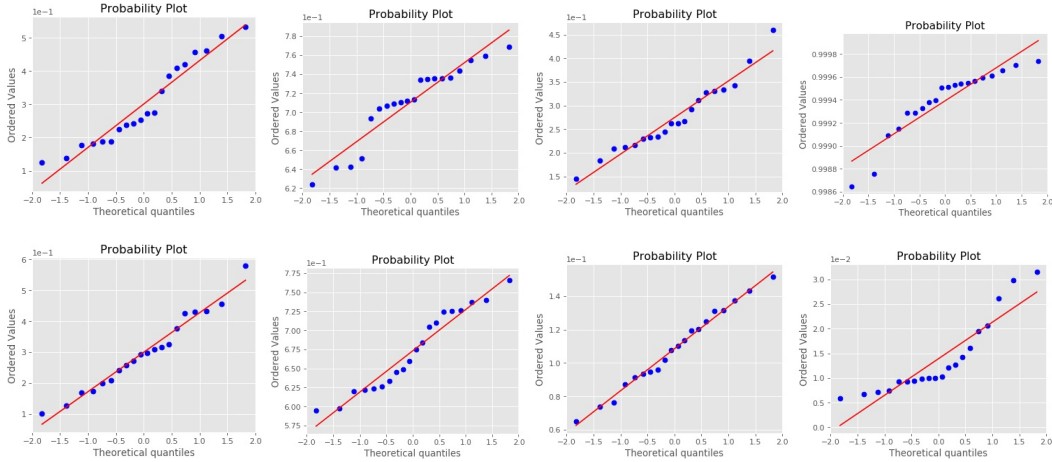

Figure 8: QQ Plot for output probabilities of VQA models: top row corresponds to the probability distributions of the class with the highest probability, and the bottom row corresponds to the probability distributions of the class with the second highest probability.

---

[4]Note that we also tried a nonparametric test, Wilcoxon rank-sum test, and obtain similar results.

## F.2 BOXPLOT FOR CIFAR-10

In this section, we visualize 5 most uncertain images for each dropout (only include Bernoulli, Concrete, and Contextual Bernoulli dropout for simplicity) leading to 15 images in total. The true images with the labels are on the left side and boxplots of probability distributions of different dropouts are on the right side. All models are trained on the original CIFAR-10 dataset. Among these 15 images, we observe that contextual dropout predicts the right answer if it is certain, and it is certain and predicts the right answer on many images that MC dropout or concrete dropout is uncertain about (e.g, many images in Figure 9-10). However, MC dropout or concrete dropout is uncertain about some easy examples (images in Figures 9-10) or certain on some wrong predictions (images in Figure 11). Moreover, on an image that all three methods have high uncertainty, concrete dropout often places a higher probability on the correct answer than the other two methods (images in Figure 11).

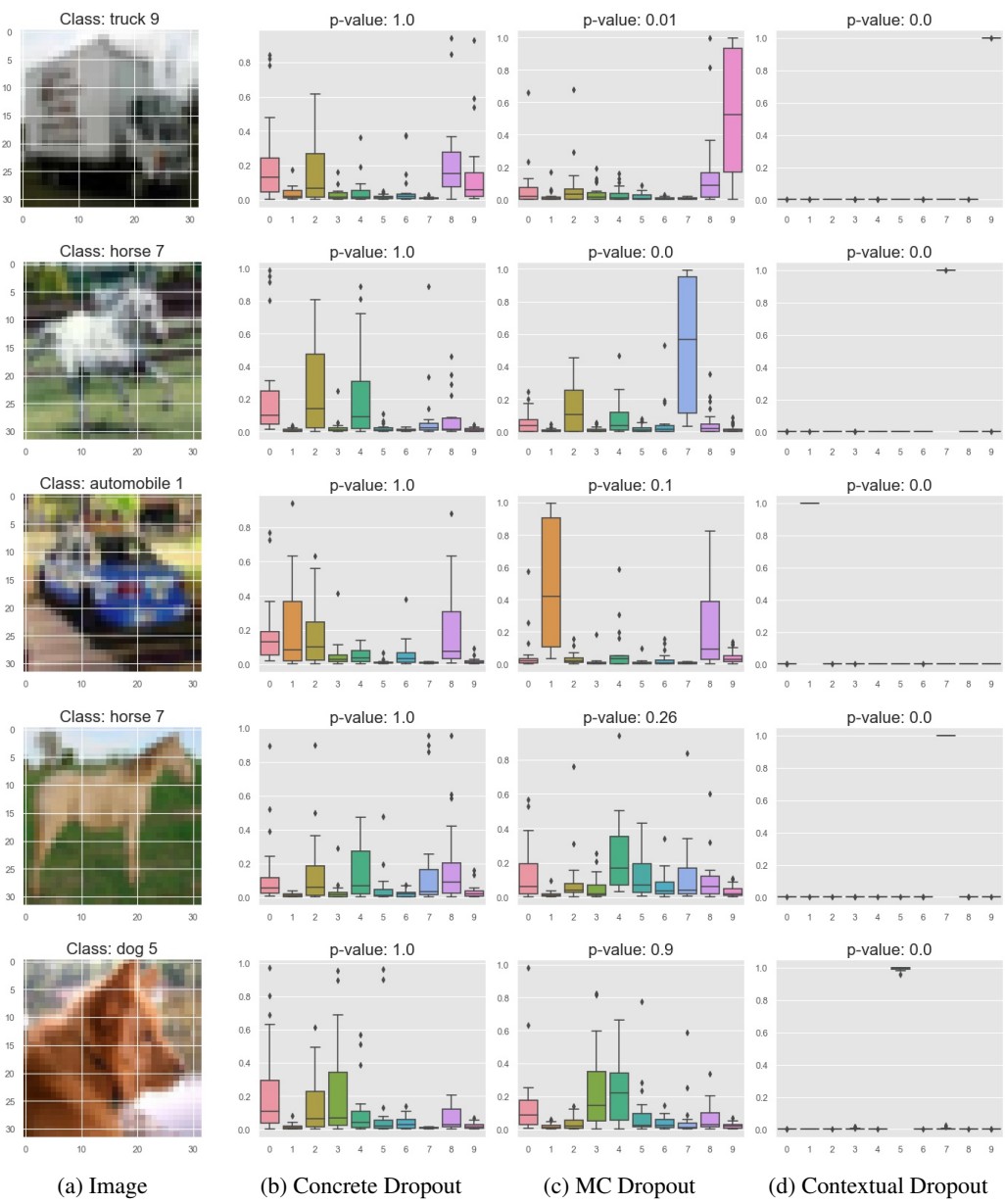

Figure 9: Visualization of probability outputs of different dropouts on CIFAR-10. 5 plots that **Concrete Dropout** is the most uncertain are presented. Number to class map: {0: airplane, 1: automobile, 2: bird, 3: cat, 4: deer, 5: dog, 6: frog, 7: horse, 8: ship, 9: truck.}

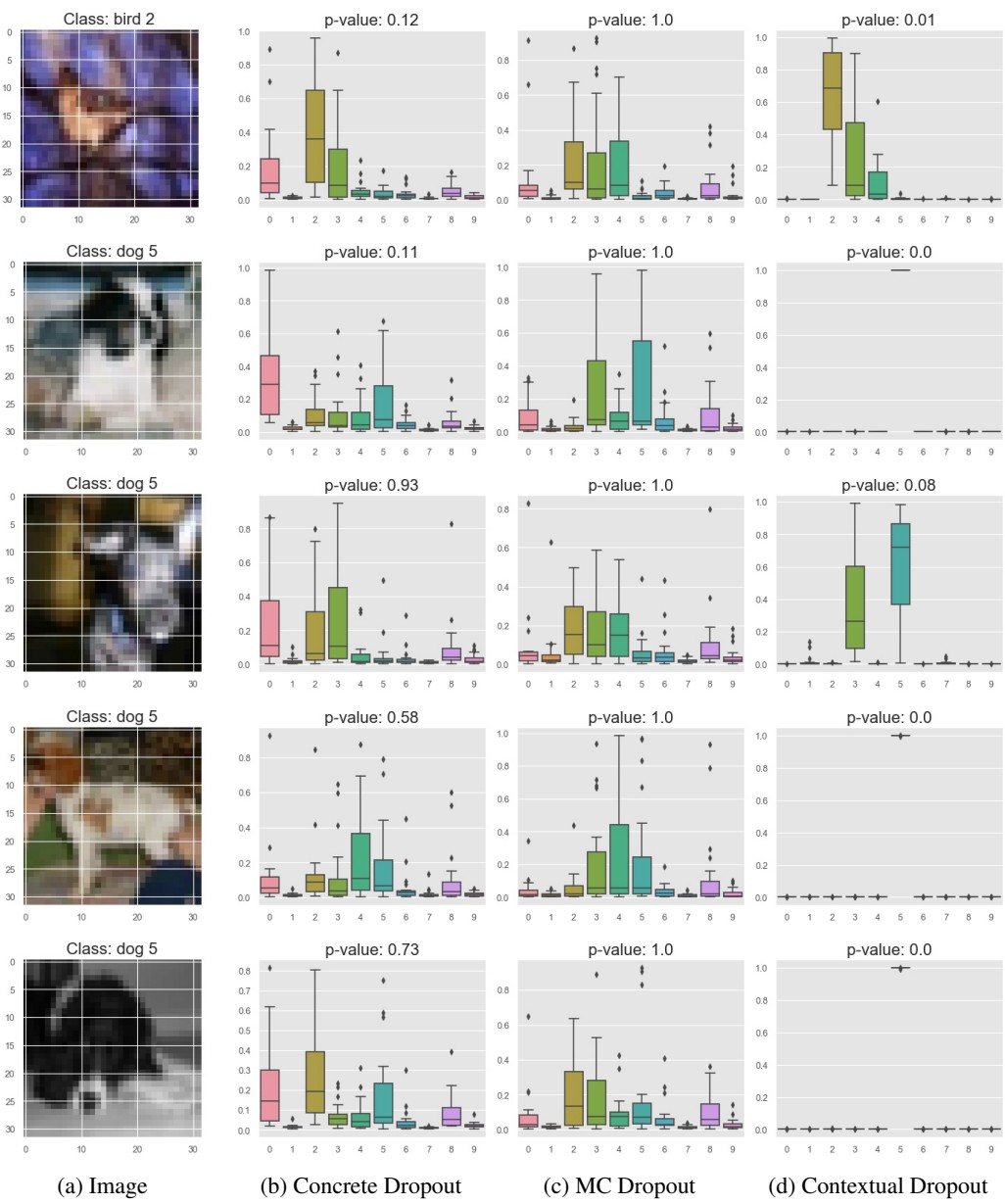

Figure 10: Visualization of probability outputs of different dropouts on CIFAR-10. 5 plots that **MC Dropout** is the most uncertain are presented. Number to class map: {0: airplane, 1: automobile, 2: bird, 3: cat, 4: deer, 5: dog, 6: frog, 7: horse, 8: ship, 9: truck.}

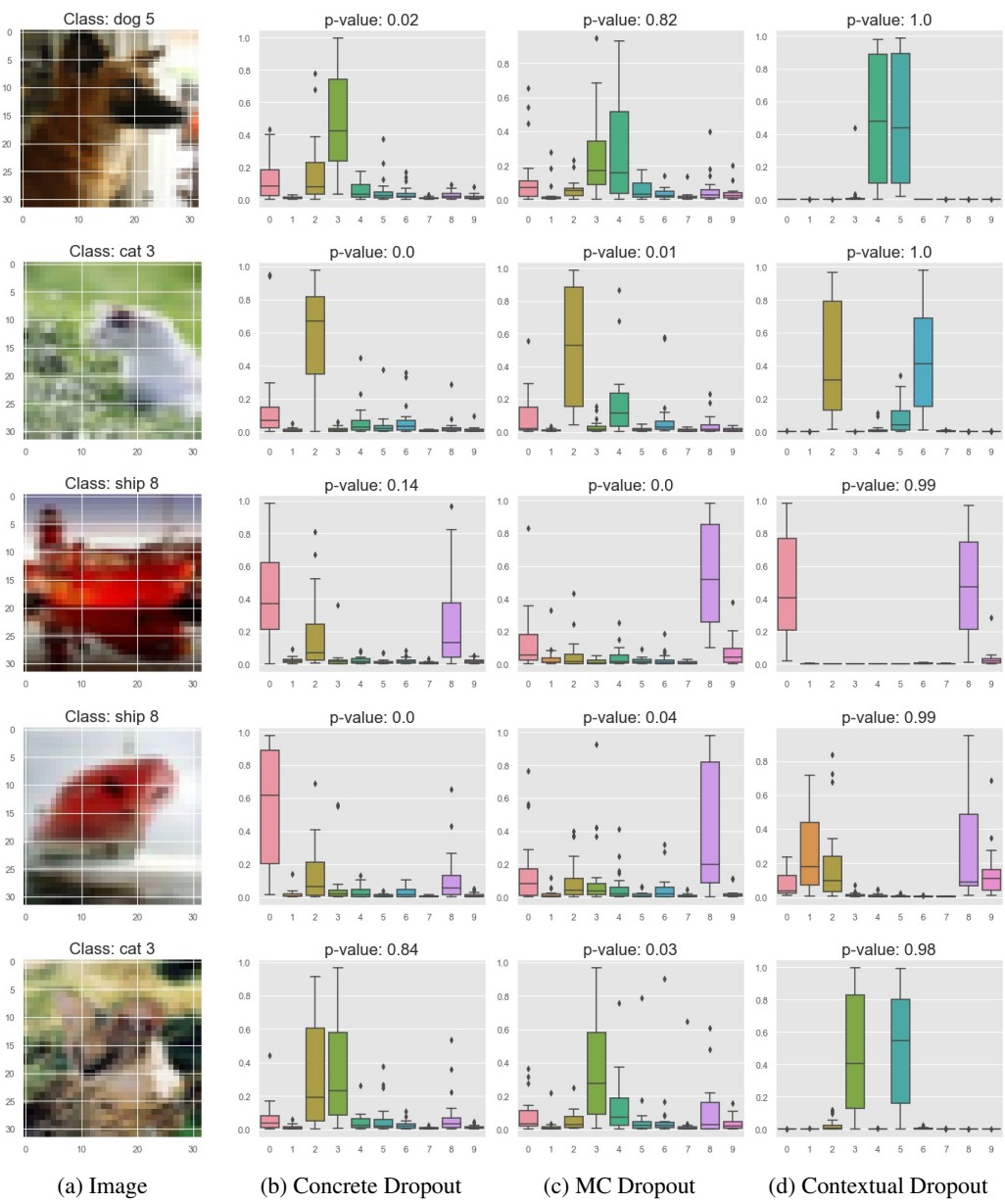

Figure 11: Visualization of probability outputs of different dropouts on CIFAR-10. 5 plots that **Contextual Dropout** is the most uncertain are presented. Number to class map: {0: airplane, 1: automobile, 2: bird, 3: cat, 4: deer, 5: dog, 6: frog, 7: horse, 8: ship, 9: truck.}

### F.3 VISUALIZATION FOR VISUAL QUESTION ANSWERING

In Figures 12-15, we visualize some image-question pairs, along with the human annotations (for simplicity, we only show the different answers in the annotation set) and compare the predictions and uncertainty estimations of different dropouts (only include Bernoulli dropout, Concrete dropout, and contextual Bernoulli dropout) on the noisy data. We include 12 randomly selected image-question pairs, and 6 most uncertain image-question pairs for each dropout as challenging samples (30 in total). For each sample, we manually rank different methods by the general rule that accurate and certain is the most preferred, followed by accurate and uncertain, inaccurate and uncertain, and then inaccurate and certain. For each image-question pair, we rank three different dropouts based on their answers and $p$-values, and highlight the best performing one, the second best, and the worst with green, yellow, and red, respectively (tied ranks are allowed). As shown in the plots, overall contextual dropout is more conservative on its wrong predictions and more certain on its correct predictions than other methods for both randomly selected images and challenging images.

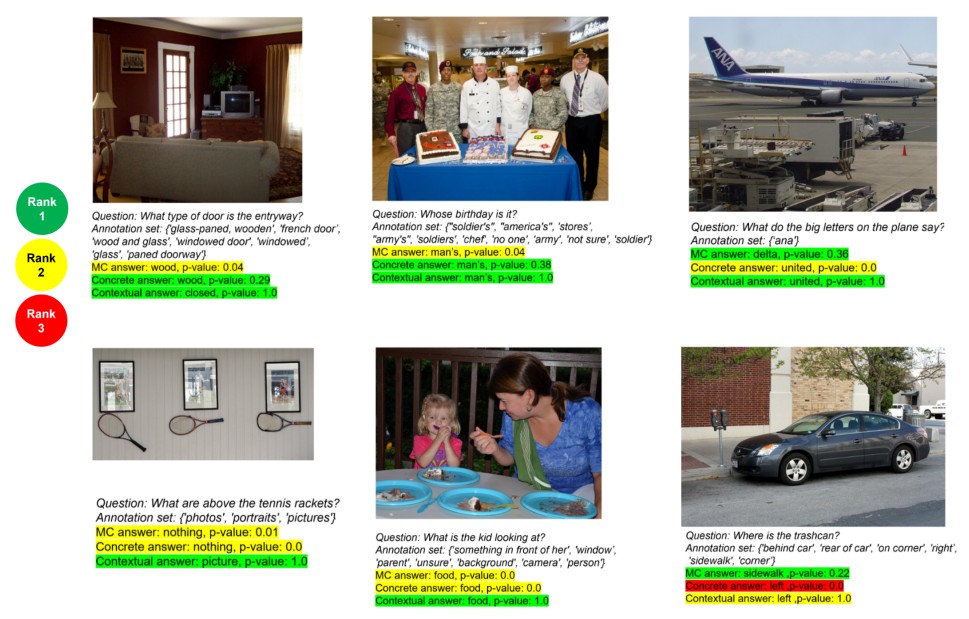

Figure 12: VQA visualization: 6 plots that **Contextual Dropout** is the most uncertain are presented.

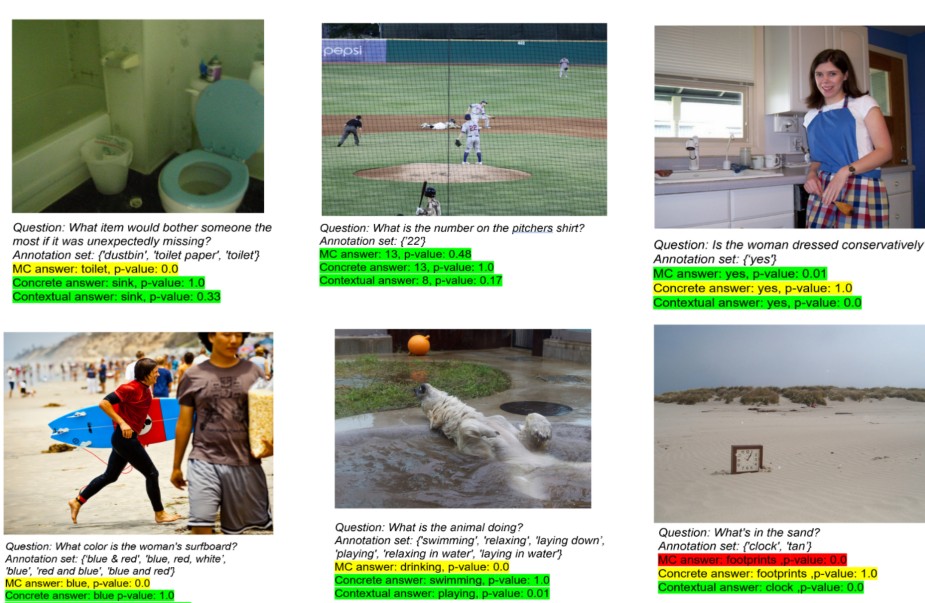

Figure 13: VQA visualization: 6 plots that **Concrete Dropout** is the most uncertain are presented.

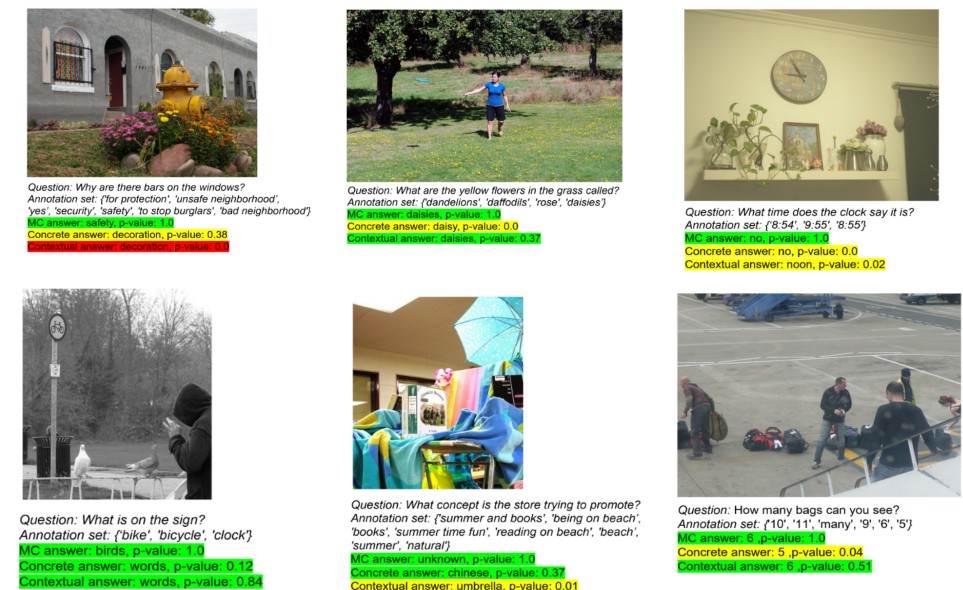

Figure 14: VQA visualization: 6 plots that **MC Dropout** is the most uncertain are presented.

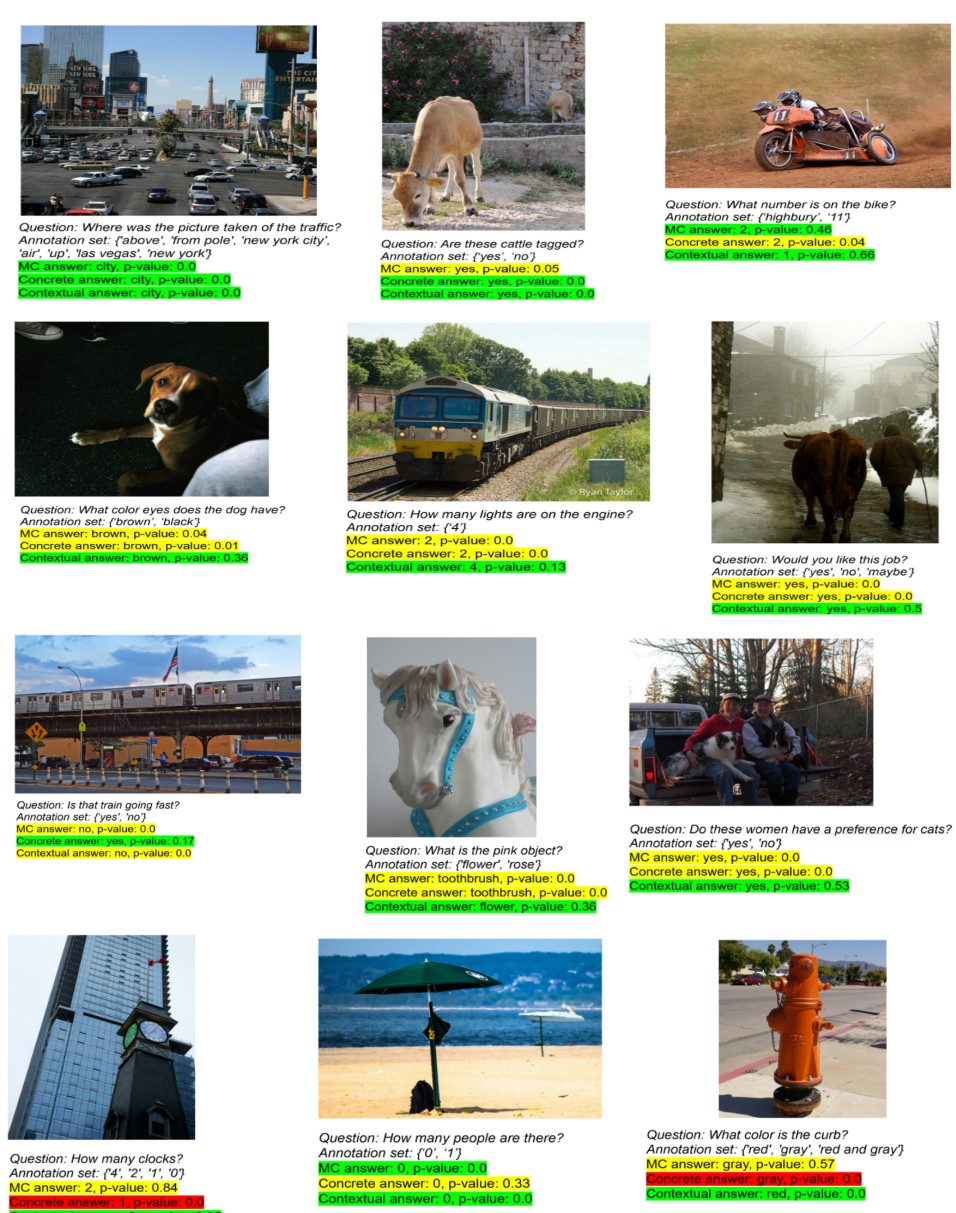

Figure 15: VQA visualization: 12 randomly selected plots are presented.

