# OpenReview forum: "Contextual Dropout: An Efficient Sample-Dependent Dropout Module"
_ICLR.cc/2021/Conference — ICLR 2021 Poster_

### Official Review · AnonReviewer2 · 2020-10-22
**Sample-dependent dropout for prediction and confidence improvements**

**Rating:** 7
**Confidence:** 4

**Review:**

Sample-dependent dropout for prediction and confidence improvements

**Quality:**

_Pros._ The authors assume that the sample-dependent dropout rate is necessary to improve model performance in aspects of prediction and confidence. In the Bayesian framework, they propose a contextual dropout module that is carefully designed considering computational efficiency since it is prone to increase model capacity and computational cost.

_Cons._ Figures and tables are intentionally resized to fit the smaller areas. Some text is hard to read. Please increase the font size of the text in the figures for a printed paper.

**Clarity:**

_Pros._ The manuscript is well-written and straightforward to understand the proposed method.

_Cons._ In Section 2.3, why do we have a multi-dimensional tensor for a fully-connected layer, instead of two-dimensional U^l? (in "Contextual dropout module for fully-connected layers" paragraph.)

**Originality:**

_Pros._ It explores the context-aware dropout method, while the other dropout-related works do not, with a few extra parameters.
Including the ImageNet and the VQA experiments is helpful to illustrate the applicability of the proposed method.
They show that the contextual dropout can successfully apply to attention networks.

_Cons._ The sample-dependent dropout has a resemblance to the squeeze-and-excitation module (the authors also mentioned in Section 2.3). In the related work section, more discussion on it would be helpful to recognize its novelty.

Sample-dependent dropout inevitably needs a sub-network to model a conditional distribution q of z for a given x. For this matter, it would be fair to compare with the squeeze-and-excitation module or self-attention models (with a small hidden dimension for parsimony; with the weights from sigmoid function).

In the proposed work, if the output of a sigmoid function is used for the real-valued weight (without the ARM estimator), instead of binary-sampled weight, does it significantly underperforms?

**What expected in rebuttal:**

(1) I would like to know that the ARM estimator in the proposed work is a critical factor, where the real-valued output of a sigmoid function is readily used for the dropout mask (as a weight). One could see this work as a variant of self-attention networks instead of a new dropout technique. In this viewpoint, the authors should defend with more persuasive logic (and experiments if possible) for their originality.

(2) For the VQA experiment, MCAN (Yu et al., 2019) gets 67.2 in the original paper with a standard dropout module with the same setting as described in the Appendix. To be fair, this report should be included. Then, the improvement from a standard dropout is reduced to 0.22 (Bernoulli contextual dropout gets 67.42 with extra parameters.)

**Minor comments:**
Footnote 1 has extra space before the period.

---

### Official Review · AnonReviewer1 · 2020-10-29
**Small but consistent improvement in experimental results.**

**Rating:** 7
**Confidence:** 4

**Review:**

Summary: This paper proposes a way of estimating data-dependent dropout probabilities. This is done in each layer using a small auxiliary neural network which takes the data (on which dropout is going to be applied) as input and outputs dropout probabilities, which are sampled and multiplied into the data.

Pros:
- The proposed model consistently leads to improvements in accuracy and uncertainty estimation over standard dropout for multiple network types and tasks (including CNNs and Transformers).
- The experimental results are thorough and include relevant p-values and error bars.
- The method is sound and well explained.

Cons:
- The gains in accuracy and uncertainty estimation for most tasks are small.
- A concern with learning the dropout probabilities on the training set is that the optimal value of dropout is zero, since that would remove regularization, thereby minimizing the training loss. It is not clear how this is avoided in the proposed approach. One thing that could prevent the dropout rates from going to zero is the prior ((\ \eta \\). However, this is also learned. So it would be good to explain what prevents the dropout rates from becoming zero.
- Conflating of 2 different effects : There are at least 2 aspects of the proposed model that could be beneficial : (1) Increase in model capacity from having a multiplicative gating interaction in the network (in expectation, the states are gated by \\(\sigma(\alpha^l)\\)), and (2) a decrease in model capacity (regularization) due to dropout noise.   An ablation analysis can help tease apart the benefits coming from these two sources.  This could be done, for example, by removing stochastic sampling (so that only the gating aspect remains) and optionally adding a regular dropout layer on the gated data. The increase in model capacity due to gating (aspect (1)) could partly explain why dropout rates do not become zero.

Overall, the model leads to consistent gains in performance with relatively extra low computational cost which makes this a good contribution. However, the significance is limited because the results are only moderately better.

Post rebuttal
The authors addressed the concerns around having a deterministic gating only baseline. I will increase my score to 7

---

### Official Review · AnonReviewer4 · 2020-10-30
**Contextual Dropout: An Efficient Sample-Dependent Dropout Module**

**Rating:** 6
**Confidence:** 4

**Review:**

##########################################################################

Summary:


The paper proposes contextual dropout as a sample-dependent dropout module, which can be applied to different models at the expense of marginal memory and computational overhead. The authors chose to focus on Visual Question Answering and Image classification tasks.  The results in the paper show the contextual dropbox can improve accuracy on ImageNet and VQA2.0 datasets.


##########################################################################

Reasons for score:


I vote for accepting. I like the idea of data-dependent dropbox and the authors showing its applicability to various neural network layers like fully connected, convolutional, and attention layers. My major concern is that it seems that the improvement seems marginal in most cases.  The clarity of the paper can also be improved. Hopefully, the authors can address my concern in the rebuttal period.


##########################################################################

Pros:


1. The paper develops a variant of dropout regularization which was shown to be effective in training deep neural nets in the past several years.

2. The proposed modified version is novel and capture data dependency more naturally relying on aleatoric uncertainty to model the uncertainty on y given x ( less explored). The computational and memory overhead is also insignificant. It is also compatible with both Bernoulli dropout and Gaussian dropout.

3.  Contextual dropout outperform alternative dropout variants and the gain is a little higher when noise is introduced



##########################################################################

Cons:


1. complexity an additional parameters even if not many might make it hard to tune. More studies on the hyper-parameter selection are needed especially based on the level of noise and the parameter

2. More experiments on a recurrent neural network model would be helpful to evaluate the generality of contextual dropout.  For example, applying contextual dropout on LSTMS / Transformer models in language tasks or vision&language tasks like image captioning.

3. contrast/comparison to Drop connect[R1], which was proposed as a generalization of dropout [R1].

[R1] Regularization of Neural Networks using DropConnect
Li Wan, Matthew Zeiler, Sixin Zhang, Yann Le Cun, Rob Fergus ; Proceedings of the 30th International Conference on Machine Learning, PMLR 28(3):1058-1066, 2013

4.

 minor
-----------
Fig5 font is not so clear and colors can be better
why gamma is between 8 and 16

---

### Comment · Area_Chair1 · 2020-11-18
**Authors: Thank you for response / Reviewers: Please update**

Thank you, authors, for your responses.

Reviewers, please read the responses and update your reviews by stating that your concerns have been addressed or by providing further rebuttal.

---

### Decision · Program_Chairs · 2021-01-07
**Final Decision**

**Decision:**

Accept (Poster)

**Comment:**

This paper proposes an input-dependent dropout strategy, using variational inference to infer the rates.  While the idea is a fairly straightforward variant of recent probabilistic dropout methods, the paper demonstrates consistent improvements across several types of NN layers (dense, convolutional, and attention) in large-scale experiments (e.g. ImageNet).  The reviewers unanimously agreed on accepting the paper.